# A photoconversion model for full spectral programming and multiplexing of optogenetic systems

Evan J Olson[1] (ID), Constantine N Tzouanas[2] & Jeffrey J Tabor[2,3,*] (ID)

## Abstract

Optogenetics combines externally applied light signals and genetically engineered photoreceptors to control cellular processes with unmatched precision. Here, we develop a mathematical model of wavelength- and intensity-dependent photoconversion, signaling, and output gene expression for our two previously engineered light-sensing *Escherichia coli* two-component systems. To parameterize the model, we develop a simple set of spectral and dynamical calibration experiments using our recent open-source "Light Plate Apparatus" device. In principle, the parameterized model should predict the gene expression response to any time-varying signal from any mixture of light sources with known spectra. We validate this capability experimentally using a suite of challenging light sources and signals very different from those used during the parameterization process. Furthermore, we use the model to compensate for significant spectral cross-reactivity inherent to the two sensors in order to develop a new method for programming two simultaneous and independent gene expression signals within the same cell. Our optogenetic multiplexing method will enable powerful new interrogations of how metabolic, signaling, and decision-making pathways integrate multiple input signals.

**Keywords** optogenetics; predictive modeling; synthetic biology; spectral multiplexing; two-component systems
**Subject Categories** Methods & Resources; Quantitative Biology & Dynamical Systems; Synthetic Biology & Biotechnology
**Mol Syst Biol. (2017) 13: 926**

## Introduction

Most optogenetic tools are based on a photoreceptor protein with a light-sensing domain that regulates an effector domain, which in turn generates a biological signal such as gene expression. One can consider a simplified model wherein a photoreceptor is produced in a "ground" state and switched to an "active" state by activating wavelengths (i.e., forward photoconversion) (Butler *et al*, 1964). Active-state photoreceptors thermally revert to the ground state with a characteristic timescale that ranges from milliseconds (Jaubert *et al*, 2007) to more than a month (Rockwell *et al*, 2012). Certain photoreceptors, exemplified by the linear tetrapyrrole (bilin)-binding phytochrome (Phy) and cyanobacteriochrome (CBCR) families, are also photoreversible where reversion from the active to ground state is driven by deactivating wavelengths (Rockwell *et al*, 2006; Möglich *et al*, 2010; Rockwell & Lagarias, 2010).

Two-component systems (TCSs) are signal transduction pathways that control gene expression and other processes in response to chemical or physical stimuli (inputs). Canonical TCSs comprise two proteins: a sensor histidine kinase (SK) and a response regulator (RR). The SK is produced in a ground state, which often (but not always) has low kinase activity toward the RR. When it detects an input via a N-terminal sensing domain, the SK uses ATP to autophosphorylate on a histidine residue within a C-terminal kinase domain. This phosphoryl group is then transferred to an aspartate on the RR. In most cases, the phosphorylated RR (RR~P) binds to a target promoter, activating transcription. Many SKs are bifunctional and the kinase domain dephosphorylates the RR~P in the absence of the input or presence of a different, deactivating input.

We have previously engineered two spectrally distinct photoreversible *Escherichia coli* TCSs, CcaSR and Cph8-OmpR (Fig EV1 and Dataset EV1) (Levskaya *et al*, 2005; Tabor *et al*, 2011; Schmidl *et al*, 2014). CcaS is a SK with a CBCR sensing domain that absorbs light via a covalently ligated phycocyanobilin (PCB) chromophore produced by an engineered metabolic pathway. Holo-CcaS is produced in an inactive, green-light-sensitive ground state, termed Pg, with low kinase activity. Upon green light exposure, CcaS Pg switches to a red-light-sensitive active state (Pr) with high kinase activity toward the RR CcaR. CcaR~P binds to the promoter $P_{cpcG2\text{-}172}$, activating transcription. Red light drives CcaS Pr to revert to Pg. Cph8 is a chimeric SK containing the PCB-binding Phy light-sensing domain of *Synechocystis* PCC6803 Cph1 and the signaling domain of *E. coli* EnvZ. In contrast to CcaS, Cph8 has high kinase activity toward the *E. coli* RR OmpR in the ground state (Pr) and

---

1 Graduate Program in Applied Physics, Rice University, Houston, TX, USA
2 Department of Bioengineering, Rice University, Houston, TX, USA
3 Department of Biosciences, Rice University, Houston, TX, USA
 *Corresponding author. Tel: +1 713 348 8316; E-mail: jeff.tabor@rice.edu

low kinase (high phosphatase) activity in a far-red absorbing activated state (Pfr). OmpR~P binds and activates transcription from the $P_{ompF146}$ promoter. Data from our group and others suggest that CcaS Pr is stable for hours or more (Hirose *et al*, 2008; Olson *et al*, 2014), while Cph8 Pfr is far less stable (Olson *et al*, 2014).

Recently, we developed a predictive phenomenological model to describe the responses of CcaSR and Cph8-OmpR to green and red light intensity signals, respectively (Olson *et al*, 2014). This model depicts a three-step response comprising a pure delay, an intensity-dependent first-order transition in output gene expression rate, and a first-order transition in the concentration of the output gene set by cell growth rate. By measuring the expression of a reporter gene over time in response to a series of light step changes of different initial and final intensities, we parameterized these three timescales for both light sensors.

Despite its predictive power, our previous model has several key limitations. First, it can only predict the responses of the optogenetic tools to the specific light sources used during parameterization. Second, it cannot account for perturbations introduced by secondary light sources such as those that might be used for simultaneous measurement of fluorescent reporter proteins or multiplexed control of both tools in the same cell. Third, the model yields few insights into the mechanistic origin of the observed response dynamics. For example, it captured, but could not elucidate the origin of, our observation that the rate of the gene expression transition depends upon the direction and final intensity of the light step change.

An *in vitro* model (i.e., of purified proteins) (Butler *et al*, 1964; Sager *et al*, 1988; Giraud *et al*, 2010) describing the intensity and wavelength dependence of switching between ground and active states has previously been used to describe photoswitching of Phys (Butler *et al*, 1964), CBCRs (Rockwell *et al*, 2012), bacteriophytochromes (Giraud *et al*, 2010), LOV domains (Swartz *et al*, 2001), and cryptochromes (Liu *et al*, 2008) among others. In this model, the sensors are characterized by their ground- and active-state photoconversion cross sections (PCSs), $\sigma_g(\lambda)$ and $\sigma_a(\lambda)$, which enable direct calculation of the forward and reverse photoconversion rates, $k_1$ and $k_2$, in response to photons of wavelength $\lambda$. Given knowledge of both PCSs ($\sigma_i(\lambda)$), one can compute both photoconversion rates ($k_i$) for a light source with a known spectral flux density $n_{light}(\lambda)$ (µmol m$^{-2}$ s$^{-1}$ nm$^{-1}$) by calculating the spectral overlap integral $k_i = \int \sigma_i \cdot n_{light} \, d\lambda$. The photoconversion rates can then be used, along with the light-independent photoreceptor "dark reversion" rate ($k_{dr}$) to calculate the populations of ground- and active-state photoreceptor.

Despite its potential for predicting photoreceptor responses to virtually any light condition, the above two-state model has not been explored for optogenetics. In particular, the complete $\sigma_i(\lambda)$ has not been determined for any optogenetic photoreceptor. While the absorbance spectrum is often well established for these sensors via *in vitro* measurement, the spectral dependence of the quantum yield (i.e., the probability of photoconversion given that a photon has been absorbed) is not. However, even if $\sigma_i(\lambda)$ were to be determined, in order to calculate ground- and active-state photoreceptor populations, the model would need to be extended to capture photoreceptor production and decay dynamics in living cells. Finally, an additional model would be needed to capture the biological events that occur downstream of the photoreceptor.

Here, we develop, experimentally parameterize, and demonstrate the predictive capabilities of an *in vivo* optogenetic TCS model. Specifically, we first extend the two-state model for the *in vivo* environment, and integrate simplified descriptions of TCS signaling and output gene expression in order to capture the complete light-to-gene-product signal transduction. Next, we develop a standard set of spectral and dynamic characterization experiments using our open-source Light Plate Apparatus (LPA) instrument (Gerhardt *et al*, 2016) enabling parameterization of the model for both CcaSR and Cph8-OmpR and estimation of $\sigma_i(\lambda)$ *in vivo*. We validate our approach by using the model to accurately predict the gene expression response of both systems to a series of spectrally and dynamically diverse light programs very different from those used for parameterization. Finally, we express CcaSR and Cph8-OmpR in the same cell and combine the models with our biological function generator approach to overcome their inherent spectral cross-reactivity and demonstrate multiplexed programming of gene expression dynamics.

## Results

### Optogenetic TCS model

We constructed an *in vivo* optogenetic TCS model comprised of a "sensing model", which converts light inputs into a ratio of the photoreceptor populations, and an "output model" which converts the photoreceptor populations into a gene expression signal. The sensing model (Materials and Methods) extends the *in vitro* two-state photoconversion model to include terms for production of new ground-state photoreceptors ($S_g$) at rate $k_S$ and dilution of both $S_g$ and active-state photoreceptors ($S_a$) at rate $k_{dil}$ to the two-state model (Fig 1A). The sensing model accepts any $n_{light}(\lambda)$ input and produces $S_g$ and $S_a$ populations as an output (Fig 1B and C). The ratio $S_a/S_g$ feeds into an "output model" comprising a phenomenological description of TCS signaling and a standard model of output gene expression (Fig 1C). The TCS signaling model (Materials and Methods) describes a pure time delay ($\tau$) and Hill function mapping $k_G(x) = \hat{b} + \hat{a} \cdot x^n/(K^n + x^n)$ between $x = S_a/S_g$ and output gene production rate ($k_G$). In our initial experiments, we utilize super-folder GFP ($G$) as the output and quantify its expression level in Molecules of Equivalent Fluorescein (MEFL) (Castillo-Hair *et al*, 2016). $\hat{a}$ is the range of possible $k_G$ values, $\hat{b}$ is the minimum value of $k_G$, $n$ is the Hill coefficient, and $K$ is the $S_a/S_g$ ratio resulting in 50% maximal system response. Together, these terms capture SK autophosphorylation, phosphotransfer, RR dimerization, DNA binding, promoter activation, and GFP production. GFP is degraded in a first-order process with rate $k_{dil}$ (Materials and Methods) and has a minimum concentration $b = \hat{b}/k_{dil}$ and concentration range $a = \hat{a}/k_{dil}$ given a constant cell growth rate.

### Light source model

Most light sources have a fixed spectral flux density (i.e., output spectrum) that scales with light intensity ($I$, µmol m$^{-2}$ s$^{-1}$). For such light sources, we can write $n_{light} = \hat{n}_{light} \cdot I$ where $\hat{n}_{light}$ is the output spectrum at 1 µmol m$^{-2}$ s$^{-1}$. To quantify the overlap between $n_{light}$ and $\sigma_i$ for a given photoreceptor, we introduce $\hat{k}_i$ as the photoconversion rate per unit light intensity (min$^{-1}$ [µmol m$^{-2}$ s$^{-1}$]$^{-1}$). Then,

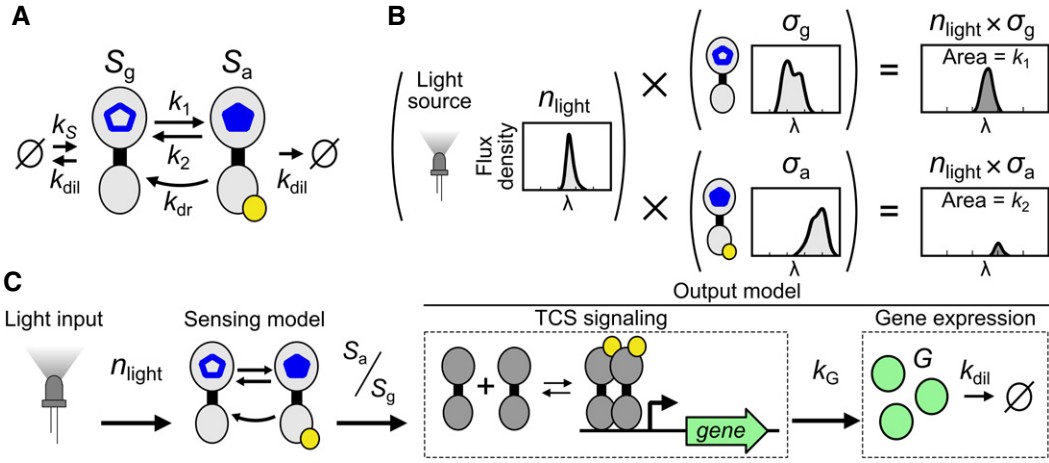

**Figure 1.   Optogenetic TCS model.**

A   The two-state photoreceptor model, which includes ground- and active-state ($S_g$ and $S_a$) photoreceptors (aka sensors), photoconversion rates $k_1$ and $k_2$, and dark reversion rate $k_{dr}$, is converted to a "sensing model" for *in vivo* environments by adding a $S_g$ production rate $k_S$ that captures both gene expression and holo-protein formation, and a dilution rate $k_{dil}$ for both $S_a$ and $S_g$ due to cell growth and sensor degradation (Materials and Methods). The hollow blue pentagon represents a chromophore in the ground state, while the filled blue pentagon represents that in the activated state.

B   Photoconversion rates are determined by the overlap integral of the spectral flux density of the light source ($n_{light}$) and the $S_g$ and $S_a$ photoconversion cross sections $\sigma_g$ and $\sigma_a$ (Materials and Methods).

C   The sensing model converts $n_{light}$ into the active ratio of light sensors $S_a/S_g$ which feeds into an "output model" with a simplified model of TCS signaling that regulates the production rate $k_G$ of the target protein $G$, which is diluted due to cell growth and proteolysis at rate $k_{dil}$ (Materials and Methods).

for a given light source, $k_i = I \cdot \int \sigma_i \cdot \hat{n}_{light}\, d\lambda = I \cdot \hat{k}_i$. That is, $k_1$ and $k_2$ take on values proportional to light intensity.

### Dynamical and spectral characterization of CcaSR

We designed a set of four simple gene expression characterization experiments to train the optogenetic TCS model for CcaSR (Fig 2A–E, Dataset EV2, and Appendix Methods S1 and S2). First, we quantify activation dynamics by preconditioning *E. coli* expressing CcaSR in the dark, introducing step increases in green light (centroid wavelength $\lambda_c$ = 526 nm, (Tables EV1–EV3, Dataset EV3, and Appendix Method S3) to different intensities, and measuring sfGFP levels over time by flow cytometry (Materials and Methods, Fig 2B, and Appendix Fig S1). Second, we measure deactivation dynamics by preconditioning the cells in different intensities of green light and measuring the response to step decreases to dark (Fig 2C and Appendix Fig S1). Third, we measure the ground-state spectral response by exposing the bacteria to 23 LEDs with $\lambda_c$ spanning 369–958 nm with illumination intensities varying over three orders of magnitude (Materials and Methods, Fig 2D, Appendix Fig S2, Tables EV1–EV3, and Appendix Method S3) and measuring sfGFP at steady state. Finally, we measure the activated state spectral response by repeating the previous experiment in the presence of a constant intensity of activating light (Fig 2E and Appendix Fig S2).

### CcaSR model parameterization

Next, we used nonlinear regression to fit the model to the dynamical and spectral characterization data (Materials and Methods, Table EV4, and Dataset EV2). Specifically, we determined $\hat{k}_1$ and $\hat{k}_2$ values for each LED, and (LED-independent) values of the Hill

function parameters, $k_{dil,}$ $k_{dr}$, and $\tau$ for the system (Fig 2F and G). While simulations using the resulting best-fit parameters (Fig 2B–E, Dataset EV4, and Table EV4) recapitulate the known properties of the system (Appendix Fig S3), the value of the Hill parameter $K$ is weakly determined. In particular, alterations in $K$ from the best-fit value can be compensated for by changes in $\hat{k}_1$ and $\hat{k}_2$ (Appendix Fig S4). Thus, we cannot confidently determine the absolute rates of forward and reverse photoconversion. Nonetheless, fixing $K$ at its best-fit value results in model predictions that quantitatively agree with the experimental measurements (Fig 2B–E and Appendix Fig S3).

### Spectral validation of the CcaSR photoconversion model

Our parameterization experiments yield $\hat{k}_1$ and $\hat{k}_2$ values for each calibration LED (Fig 2G). However, to predict the response of an optogenetic tool to a new light source without additional calibration experiments, knowledge of $\sigma_i$ is required. To estimate $\sigma_i$ for CcaSR, we developed a procedure to fit a cubic spline to the previously determined $\hat{k}_1$ and $\hat{k}_2$ values for each of the 23 LEDs (Materials and Methods, Fig 3A, Appendix Figs S5 and S6, and Datasets EV2 and EV5). Importantly, our regression procedure considers the response of CcaSR to the full spectral output of each LED, not just its centroid wavelength. To validate the resulting $\sigma_i$ estimate, we measured $\hat{n}_{light}(\lambda)$ for a previously untested set of eight color-filtered white-light LEDs designed to have complex spectral characteristics (Tables EV1–EV3, Dataset EV3, and Appendix Method S3) and calculated an expected $\hat{k}_i$ for each (Fig 3B). In combination with the remaining model parameters (Fig 2F), we used these $\hat{k}_i$ to predict the steady-state intensity dose-response to these eight LEDs in the presence and absence of activating light ($\lambda_c$ = 526 nm). These predictions are

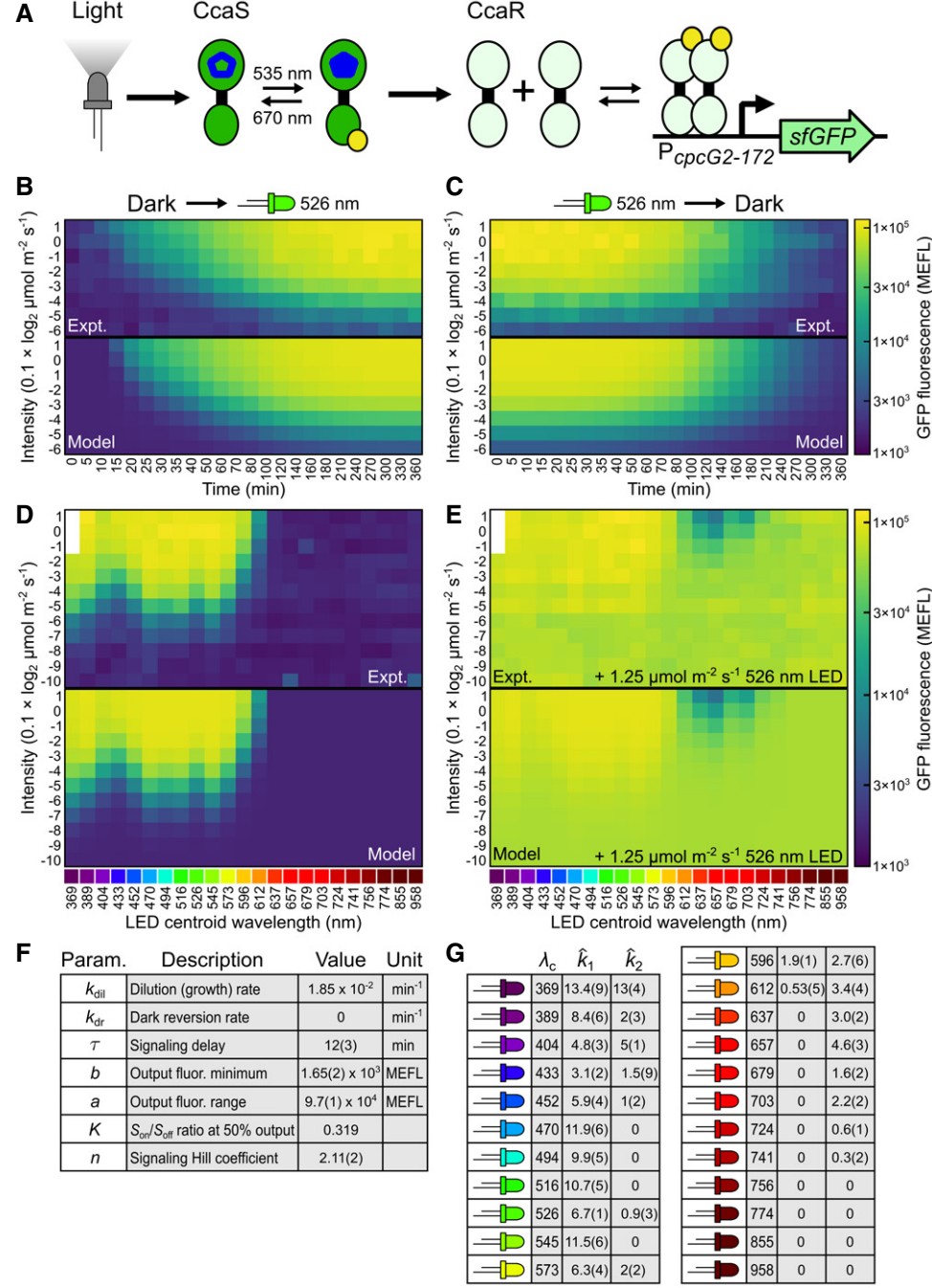

**Figure 2. Characterization and model parameterization for CcaSR.**

A    Schematic of CcaSR TCS with sfGFP output. Wavelength values represent *in vitro* measured absorbance maxima.

B–E   Training data for the full CcaSR system model (Fig 1C). Experimental observations ("Expt.") and simulations of the best-fit model ("Model") are shown for each set. In particular, the response dynamics to step (B) increases from dark to eight different intensities and (C) decreases from eight different intensities to dark were evaluated using the $\lambda_c$ = 526 nm LED. Time points are distributed unevenly to increase resolution of the initial response. (D, E) Steady-state intensity dose–response to a set of 23 "spectral LEDs" with $\lambda_c$ spanning 369 nm to 958 nm. (D) Forward photoconversion is primarily determined by the response to the spectral LEDs. (E) Reverse photoconversion is analyzed by including light from a second, activating LED ($\lambda_c$ = 526 nm at 1.25 μmol m$^{-2}$ s$^{-1}$). The $\lambda_c$ = 369 nm LED is not capable of reaching the brightest intensities, and thus, those data points are not included. Light intensities are shown in units of 0.1 × log$_2$ μmol m$^{-2}$ s$^{-1}$ scale (e.g., a value of 1 corresponds to 10 × 2$^1$ = 20 μmol m$^{-2}$ s$^{-1}$). sfGFP fluorescence is calibrated to MEFL units (Materials and Methods). Each row of measurements in panels (B–E) was collected in a single 24-well plate. The 40 plates required to produce the training dataset were randomly distributed across eight LPAs over five separate trials (Materials and Methods and Dataset EV2). Each color patch represents the arithmetic mean of a single population of cells.

F, G   Best-fit model parameters produced via nonlinear regression of the model to training data (Materials and Methods and Table EV4). $\hat{k}_i$ are unit photoconversion rates ($10^{-3}$ × min$^{-1}$/(μmol m$^{-2}$ s$^{-1}$), that is, $k_i = I \cdot \hat{k}_i$, where $I$ is the LED intensity in μmol m$^{-2}$ s$^{-1}$). Uncertainty in the least-significant digits are indicated in parenthesis.

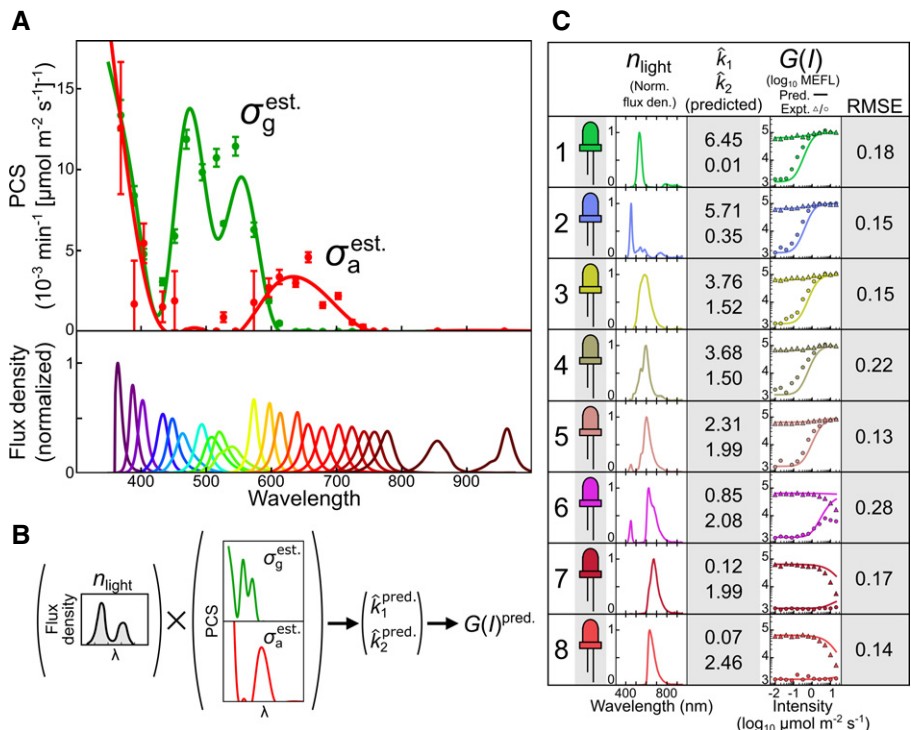

**Figure 3. Estimation of the CcaS photoconversion cross section and spectral validation of the CcaSR model.**

A   We estimate the continuous ground- and active-state PCSs of CcaS ($\sigma_i^{est.}$, lines) by regressing cubic splines to minimize the difference between experimentally determined photoconversion rates (points) and those predicted via $\hat{k}_i^{pred.} = \int \sigma_i^{est.} \cdot \hat{n}_{light} \, d\lambda$ (Materials and Methods, Appendix Figs S5 and S6, and Dataset EV5). Error bars indicate the standard error of the best-fit values of the photoconversion rates that were determined during model parameterization of CcaSR (Fig 2). The normalized spectral flux densities of the spectral LEDs are shown at bottom.

B   Using $\sigma_i^{est.}$ to predict photoconversion rates for light sources not in the spectral LED training set. Predicted photoconversion rates are integrated into the CcaSR model by keeping all other parameters (Fig 2F) fixed, enabling prediction of the intensity dose-response of CcaSR to the new light source (i.e., $G(I)^{pred.}$).

C   Spectral validation of the CcaSR model and $\sigma_i^{est.}$ consists of prediction of the intensity dose-response for eight challenging, broad-spectrum light sources constructed by applying colored filters over white-light LEDs (Materials and Methods, Tables EV1–EV3, and Dataset EV3). Measured $n_{light}$, predicted $\hat{k}_i$ ($10^{-3} \times \min^{-1}/(\mu\mathrm{mol}\ \mathrm{m}^{-2}\ \mathrm{s}^{-1})$), measured and predicted intensity dose-response curves, and RMSE between model and prediction are shown for each LED (Materials and Methods). The forward and reverse intensity responses are determined using the filtered LED alone (circles) and in the presence of a second activating LED ($\lambda_c$ = 526 nm at 1.25 $\mu\mathrm{mol}\ \mathrm{m}^{-2}\ \mathrm{s}^{-1}$, triangles). The simulated responses are determined using the calculated photoconversion rates (Materials and Methods). RMSE relative errors are expressed in $\log_{10}$ decades (Materials and Methods). Data were collected across four LPAs, and the forward (circles) and reverse (triangles) intensity responses were collected over two separate experimental trials (Materials and Methods and Dataset EV2). Each data point represents the arithmetic mean of a single population of cells.

remarkably accurate for LEDs 1–5 (root-mean-square errors (RMSEs) from 0.13 to 0.22, Materials and Methods), which drive sfGFP to high levels, and 7 and 8, which drive low expression (RMSE = 0.17 and 0.14, respectively), but slightly less so for LED 6 (RMSE = 0.28), which drives sfGFP to an intermediate expression level (Fig 3C). These results demonstrate that we can predict the response of CcaSR to a wide range of previously untested light sources using only spectroradiometric measurements of their $\hat{n}_{light}(\lambda)$ and not biological calibration experiments.

**Dynamic validation of the CcaSR photoconversion model**

We previously developed a "biological function generator" method in which we use a predictive model to computationally optimize light input programs to drive tailor-made gene expression signals such as linear ramps and sine waves (Olson *et al*, 2014). This method constitutes a rigorous validation of the predictive power of a model because the light inputs and gene expression outputs are temporally complex and cover a wide range of levels. To validate

our CcaSR photoconversion model, we first designed a challenging reference gene expression signal (Fig 4 and Dataset EV6). The signal starts at *b* and then increases linearly (on a logarithmic scale) over 90% of the total CcaSR response range over 210 min. After a 60-min hold, the signal decreases linearly to an intermediate expression level over another 210 min. We then used the model to computationally design four light time courses each with different LEDs or LED mixtures to program the bacteria to follow this reference signal (Materials and Methods and Dataset EV6). "UV mono" utilizes a single UV LED ($\lambda_c$ = 389 nm) (Fig 4A) to demonstrate control of CcaSR with an atypical light source. "Green mono" uses the $\lambda_c$ = 526 nm LED (Fig 4B) to demonstrate predictive control with a typical light source. "Red perturbation" combines "Green mono" with a strong red ($\lambda_c$ = 657 nm) sinusoidal signal (Fig 4C and Dataset EV6) designed to demonstrate the perturbative effects of additional sources of light during experiments. Finally, in "Red compensation", the "Green mono" time course is re-optimized to compensate for the impact of "Red perturbation" (Fig 4D and Materials and Methods).

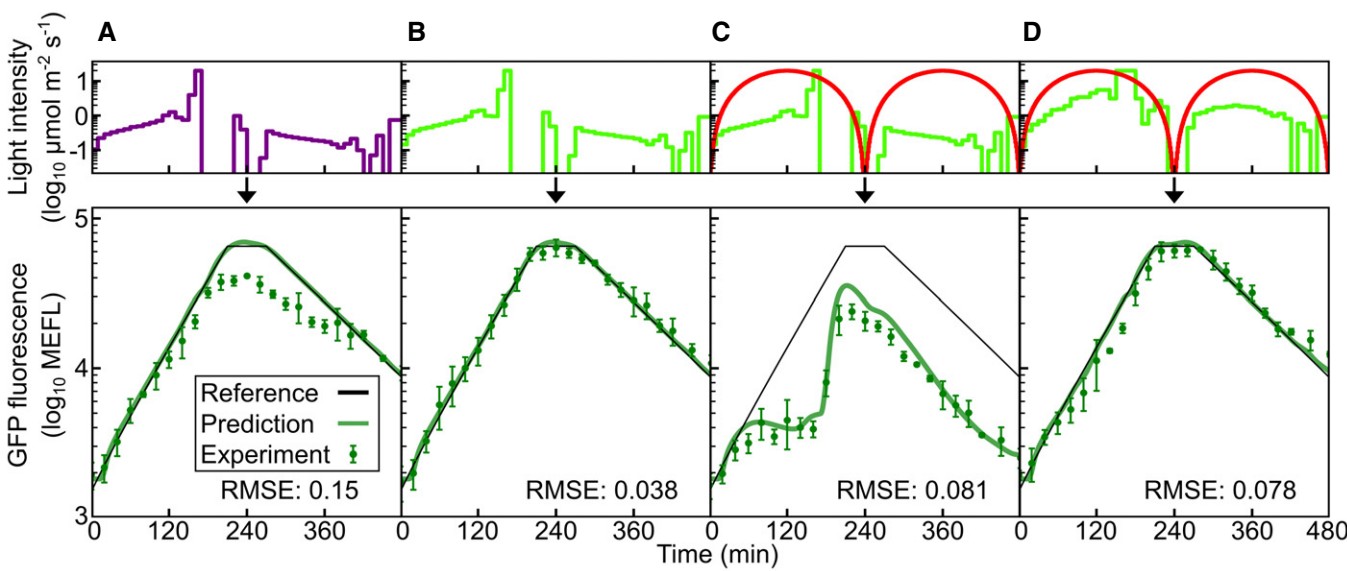

**Figure 4. Dynamical validation of the CcaSR model.**

We compare model predictions of dynamical CcaSR sfGFP output to experimental measurements for time-varying light inputs from UV (purple line; $\lambda_c$ = 389 nm), green ($\lambda_c$ = 526 nm), or green plus red ($\lambda_c$ = 657 nm) light. In all cases, the light programs (top) are produced using the light program generator algorithm (LPG, Materials and Methods). The LPG uses the model of the system to produce a light program that drives a gene expression simulation (bottom, green line) which closely matches the reference signal (bottom, black line). The simulation (i.e., model prediction), is then compared to the experimentally measured response (bottom, data points). The reference signal consists of a ramp up, hold, and ramp down on a logarithmic scale (Dataset EV6).

A  "UV mono". The LPG-generated UV light signal drives the CcaSR system along a trajectory predicted to follow the reference signal.

B  "Green mono". The green LED alone provides an optimized input signal.

C  "Red perturbation". The green LED provides the "Green mono" signal, while the red LED generates a sinusoidal perturbative signal (center) with a 240-min period and 20 μmol m⁻² s⁻¹ peak-to-peak amplitude.

D  "Red compensation". The red perturbative signal is again present. However, the LPG redesigns the green light signal to account for its presence.

Data information: Light signals are shown in units of $\log_{10}$ μmol m⁻² s⁻¹, and RMSE relative errors are expressed in $\log_{10}$ decades (Materials and Methods). Error bars correspond to the standard deviation in fluorescence measurements over three independent experimental trials (Table EV4 and Dataset EV2).

The model predicts the response of CcaSR to all four light signals with high quantitative accuracy (Fig 4 and Dataset EV2). "Mono UV" presents the greatest challenge, resulting in an RMSE of 0.15 (Fig 4A). We suspect that prediction errors in this program are due to PCB photodegradation, as we observed no significant toxicity via bacterial growth rate during this experiment (Appendix Figs S7 and S8), and the prediction remains accurate until UV reaches maximum intensity (20 μmol m⁻² s⁻¹). "Green mono" (Fig 4B) results in the lowest error (RMSE = 0.038), which is expected because this LED was used to perform the dynamic calibrations (Fig 2B and C). As intended, "Red perturbation" results in an enormous deviation from the reference signal (Fig 4C), and the model accurately predicts this effect (RMSE = 0.081). Finally, "Red compensation" demonstrates that the effect of the perturbation can be eliminated using our model (Fig 4D, RMSE = 0.078).

### Cph8-OmpR photoconversion model

To evaluate the generality of our approach, we repeated the entire workflow for Cph8-OmpR (Figs EV2–EV4, Appendix Figs S9–S11, Table EV5, and Dataset EV7). Though CcaSR and Cph8-OmpR are both photoreversible TCSs, they have different photosensory domains, ground-state activities, and dynamics. To account for the fact that Cph8-OmpR is produced in an active ground state, we used

a repressing Hill function in the TCS signaling portion of the output model (Materials and Methods). The model again fits exceptionally well to the experimental data (Fig EV2 and Appendix Figs S9–S11). Unlike CcaSR, which exhibited no detectable dark reversion (Fig 2F), Cph8-OmpR appears to revert in $\tau_{1/2} = \ln 2 / k_{dr} = 5.5$ min (Fig EV2F). As before, $K$ is underdetermined (Appendix Fig S4), and we chose the best-fit value (Table EV5). The Cph8-OmpR model performs similarly to its CcaSR counterpart in the spectral validation experiments (Fig EV3) and demonstrates greater predictive control in the dynamic validation experiments (Fig EV4).

### Development of a CcaSR, Cph8-OmpR dual-system model

We engineered a three-plasmid system (Fig EV1 and Dataset EV1) to express CcaSR and Cph8-OmpR in the same cell with sfGFP and mCherry outputs, respectively (Fig 5A). To recalibrate for mCherry [quantified in Molecules of Equivalent Cy5 (MECY)] and any changes due to the new cellular context, we measured the steady-state levels of sfGFP and mCherry at different combinations of green ($\lambda_c$ = 526) and red ($\lambda_c$ = 657) light (Fig 5B, Appendix Fig S12, and Dataset EV8) and refit the Hill function parameters of the TCS signaling portion of the output model (Table EV6). Because the photoconversion parameters are properties of the photoreceptors themselves, we left them unchanged. The dual-system model

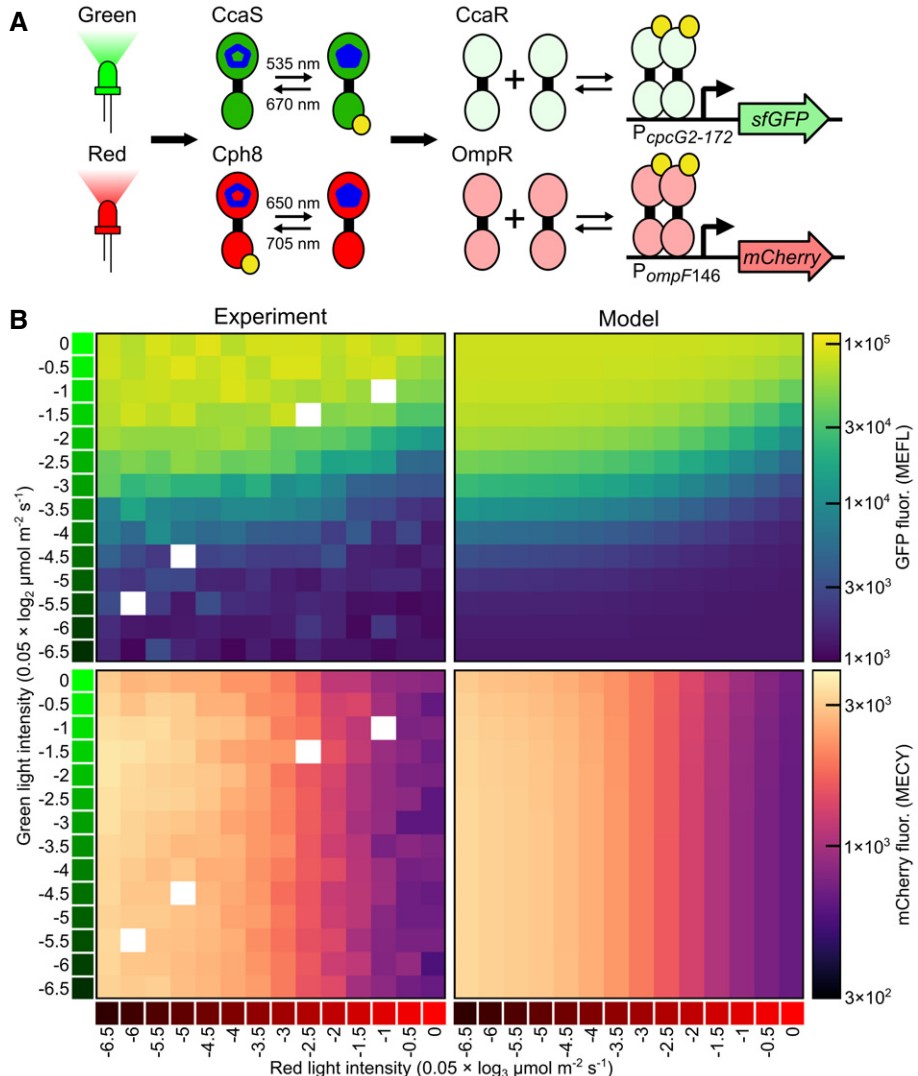

**Figure 5. Characterization and modeling of a multiplexed CcaSR/Cph8-OmpR system.**

A  CcaSR and Cph8-OmpR are co-expressed in a single strain. CcaSR regulates the expression of sfGFP, while Cph8-OmpR regulates the expression of mCherry. Wavelength values are as in Fig 2A.

B  Training data for the multiplexed model ("Experiment", Dataset EV8) consists of a two-dimensional steady-state intensity dose-response to green ($\lambda_c$ = 526 nm) and red ($\lambda_c$ = 657 nm) light. The light intensities are logarithmically distributed, with the green light varying on a 0.05 × $\log_2$ μmol m$^{-2}$ s$^{-1}$ scale (e.g., a value of −1 corresponds to 20 × 2$^{-1}$ = 10 μmol m$^{-2}$ s$^{-1}$) and the red light varying over a 0.05 × $\log_3$ μmol m$^{-2}$ s$^{-1}$ scale (e.g., a value of −1 corresponds to 20 × 3$^{-1}$ = 6.67 μmol m$^{-2}$ s$^{-1}$). The different intensity ranges are used to maintain a high-resolution measurement despite the differences in the intensity dose-responses of the two systems. The four missing intensity values (white boxes) were not collected. The training data were used to re-fit the *a*, *b*, *n*, and *K* Hill function parameters for the CcaSR and Cph8-OmpR models (Table EV6). Simulated steady-state responses to the same light environments for the best-fit dual-system models (Table EV6) are shown ("Model"). mCherry fluorescence is calibrated to MECY units (Molecules of Equivalent Cy5, Materials and Methods). RMSE relative errors are expressed in $\log_{10}$ decades (Materials and Methods). Data were collected in one experimental trial, and the 192 samples were randomly distributed across eight LPAs (Materials and Methods, Table EV6, and Dataset EV8). Each color patch represents the arithmetic mean of a single population of cells.

accurately captures the experimental observations from the characterization dataset (Fig 5B).

To validate the dual-system model, we again used the biological function generator approach (Fig 6 and Dataset EV8). We designed a series of four dual sfGFP/mCherry expression programs to increasingly challenge the model: "Green mono" using only green light and intended only to control CcaSR (Fig 6A), "Red mono" using only red light and intended to control only Cph8-OmpR (Fig 6B), "Sum", a simple combination of the first two programs (Fig 6C), and

"Compensated sum" where the green light time course is re-optimized to account for the presence of the red signal (Fig 6D) as before (Materials and Methods). Due to the minimal response of dual-system Cph8-OmpR to green light (Fig 5B), there was no need to adjust the red program to compensate for the presence of green light. The validation experimental results (Fig 6) show that our dual-system model accurately captures both sfGFP and mCherry expression dynamics. The CcaSR predictions are nearly as accurate as the single-system experiments (Fig 4), and the Cph8-OmpR

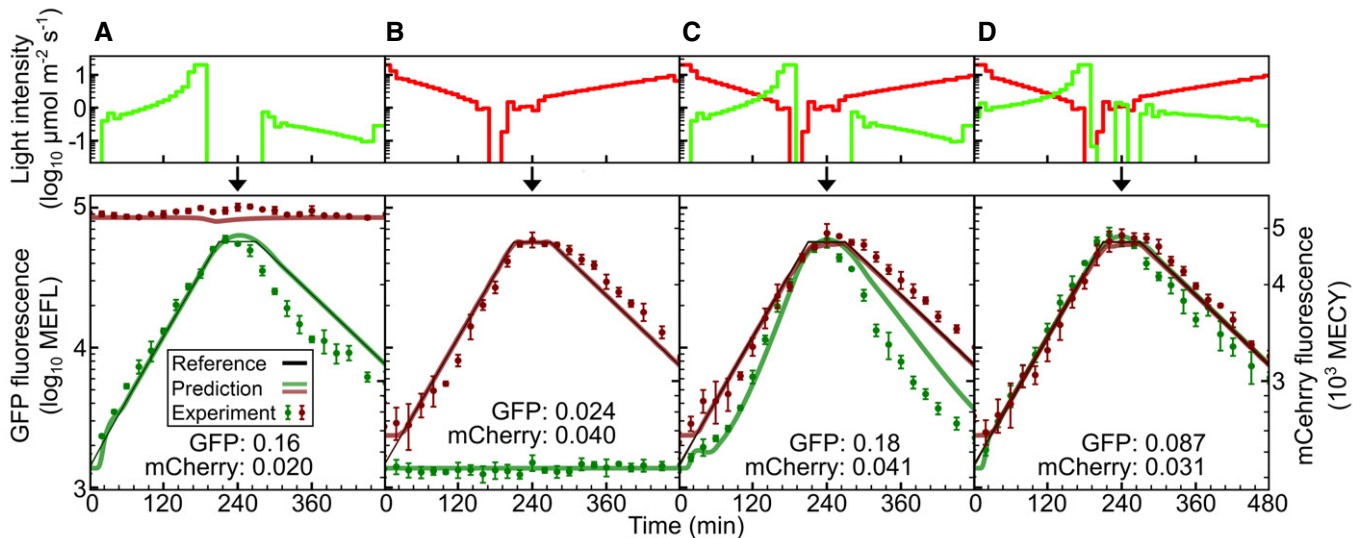

**Figure 6.** Validation of the multiplexed system model.

Predicted responses of the multiplexed system (Fig 5A) to time-varying signals of green ($\lambda_c$ = 526 nm) and red ($\lambda_c$ = 657 nm) light are compared to experimental results. Reference signals, light programs, and experimental data are as in Fig 4.

A    "Green mono". The green LED alone provides an optimized input signal for CcaSR.

B    "Red mono". The red LED alone provides an optimized input for Cph8-OmpR.

C    "Sum". The "Green mono" and "Red mono" programs are used simultaneously without any compensation, leading to a substantial deviation of the CcaSR output from the reference trajectory.

D    "Compensated sum". The "Red mono" program is used; however, the green light program is produced while incorporating red light program into the LPG (above).

Data information: RMSE relative errors are expressed in $\log_{10}$ decades (Materials and Methods). Error bars correspond to the standard deviation in fluorescence measurements over three separate experimental trials (Table EV6 and Dataset EV8).

---

results match single-system accuracy (Fig EV4), demonstrating the extensibility of our approach to multiple optogenetic tools.

### Multiplexed biological function generation

Finally, we designed and experimentally implemented four multiplexed sfGFP/mCherry expression functions representing classes of signals useful for gene circuit characterization (Datasets EV6 and EV8). "Dual-sines" illustrates that two gene expression sinusoids with different offsets, amplitudes, and periods can be composed without interference (Fig 7A). Variations of this combination of signals could be used to perform frequency analysis of multiple nodes in a gene network. "Sine and stairs" demonstrates that our approach can generate two completely different gene expression signals at the same time (Fig 7B). "Dual-stairs" demonstrates that the ratio of two proteins can be varied over a remarkably wide range (Fig 7C). Finally, "Time-shifted waveform" (Fig 7D) demonstrates that our approach can be used to characterize genetic circuits where time-delays are critical, such as those involved in cellular decision-making.

## Discussion

Our optogenetic TCS model is superior to current alternatives by several key criteria. First, like our previous version (Olson *et al*, 2014), it is quantitatively predictive and requires no parameter recalibrations from day-to-day. However, while the previous model requires experimental calibration against each light source used, the current one requires only a single set of calibration experiments and then generalizes to virtually any light source or mixture of light sources whose spectral characteristics can be measured using a spectroradiometer. Second, our optogenetic TCS model is compatible with photoreceptors with very different action spectra, opposite ground vs. active-state signaling logic, and dramatically different dark reversion timescales. Third, the current model modularly decouples the processes of sensing (photoconversion) and output (signal transduction and gene expression). The sensing model component (Fig 1A) should be compatible with a wide range of photoreceptors, including those in other organisms, because the core two-state photoswitching mechanism is used to describe their performance *in vitro*. Then, to describe optogenetic tools based upon those photoreceptors, our TCS output model can be replaced with alternatives appropriate to other pathways.

A major current problem in optogenetics is that tools developed in different studies are characterized using different culturing conditions, experiments, light sources, reporters, metrics, and so on. This lack of standardization makes it challenging to compare the performance features of different optogenetic tools on even a qualitative basis. The modeling and characterization approach we develop here is based on openly available optical hardware (Gerhardt *et al*, 2016) and flow cytometry analysis and calibration software (Castillo-Hair *et al*, 2016). Thus, our results could be directly reproduced in other laboratories using the light programs reported here (Datasets EV2, EV7 and EV8). Furthermore, our approach could be used to make data sheets that describe the behavior of diverse optogenetic tools in standard units. This benefit would enable researchers to choose the

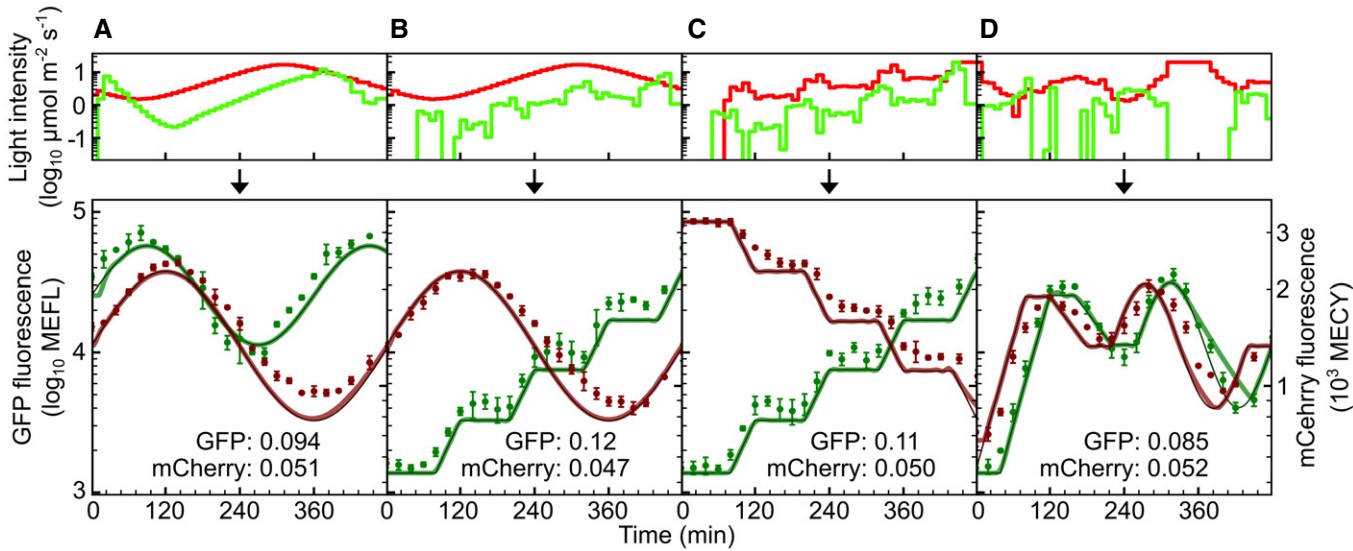

**Figure 7. Multiplexed biological function generation.**

The LPG is used to program CcaSR and Cph8-OmpR outputs to independently follow different reference signals. Red light ($\lambda_c$ = 657 nm) programs are optimized first using the LPG, and then, the "Compensated" approach (Fig 6D) is utilized to generate the green light ($\lambda_c$ = 526 nm) program (Materials and Methods).

A    "Dual-sines". The sfGFP and mCherry reference trajectories are sinusoids with different periods, amplitudes, and offsets.

B    "Sine and stairs". The mCherry signal follows the same sinusoid in "Dual-sines", but the sfGFP reference is a stepped trajectory with several plateaus and increasing linear ramps.

C    "Dual-stairs". The sfGFP signal follows the same stair-shape in "Sine and stairs"; however, the mCherry response is a decreasing stair-shape.

D    "Time-shifted waveform". The sfGFP and mCherry reference trajectories both follow the same arbitrary waveform consisting of ramps, holds, and a sinusoid, with sfGFP trailing mCherry by 40 min.

Data information: RMSE relative errors are expressed in $\log_{10}$ decades (Materials and Methods). Error bars correspond to the standard deviation in fluorescence measurements over three independent experimental trials (Table EV6 and Dataset EV8).

most appropriate tool for different applications. Additionally, short-comings of specific tools could be identified, informing efforts to optimize performance by rational approaches such as protein design (McIsaac *et al*, 2014; Engqvist *et al*, 2015; Guntas *et al*, 2015).

Our approach should enable better control of optogenetic tools with alternative or highly constrained optical hardware used in many research laboratories. For example, many groups perform single-cell optogenetic studies using fluorescence microscopes with severely restricted optical configurations. Alternatively, consumer projectors or tablet displays are potentially powerful, low-cost hardware options for optogenetics (Stirman *et al*, 2012; Beyer *et al*, 2015). The output spectrum of the light source can be measured and integrated into our workflow. After a simple recalibration (e.g., Fig 5) to account for any changes due to the new growth environment, one should be able to predict and control the optogenetic tool using the new light source.

Oftentimes, it is desirable to simultaneously control an optogenetic tool while imaging a cell of interest using white-light sources and excitation light for fluorescent reporters. Such alternative sources of illumination can have deleterious effects on the ability to control the optogenetic tool. However, if the nature of the alternative light signal is known, our approach can compensate for such perturbations (e.g., Figs 6 and 7). *In silico* feedback control has also been used to drive desired gene expression dynamics in optogenetic experiments (Milias-Argeitis *et al*, 2011, 2016; Melendez *et al*, 2014). The major benefit of this approach is that perturbations of unknown origin can be compensated by monitoring deviations in

the output of an optogenetic tool relative to a reference. Our model is compatible with such *in silico* feedback control methods.

While basic multichromatic control of optogenetic tools has been previously demonstrated (Tabor *et al*, 2011; Müller *et al*, 2013), the multiplexed biological function generation approach demonstrated here dramatically extends the capabilities of these systems, enabling implementation of several classes of experiments. We have previously shown that expression dynamics of transcription factors, as well as fluorescent proteins, can be controlled with our optogenetic tools (Olson *et al*, 2014). First, the two-dimensional response of a genetic circuit or signaling pathway could be rapidly evaluated with high reproducibility and precision. For example, one could map the response of two-input transcriptional logic gates (Nielsen *et al*, 2016), which integrate the expression levels of two different transcription factors by systematically and independently varying their expression levels while measuring the gate output with a reporter gene. The dynamics of such gates are otherwise difficult to evaluate and seldom characterized (Olson & Tabor, 2014). Second, the input/output dynamics of a transcriptional circuit could be characterized as a function of the state of the circuit itself. For example, one could evaluate how well a synthetic transcriptional oscillator can be entrained (Stricker *et al*, 2008; Mondragón-Palomino *et al*, 2011) as a function of the strength of a feedback node. In this case, one optogenetic tool could be used for the entrainment, while the second was used to alter expression level of a circuit transcription factor regulating feedback strength. Third, transcription and proteolysis (Fernandez-Rodriguez & Voigt, 2016) could be independently

controlled with two different optogenetic tools to alternatively program rapid increases or decreases in expression level. Such an approach could accelerate the gene expression signals that we have generated in this and our previous study (Olson *et al*, 2014), enabling characterization of gene circuit dynamics on faster time-scales. Finally, multiplexed biological function generation could be used to evaluate how the timing of expression of two genes impacts cellular decision-making (Kuchina *et al*, 2011; Vishnoi *et al*, 2013; Castillo-Hair *et al*, 2015). For example, in *Bacillus subtilis*, the gene circuits that regulate sporulation and competence compete via a "molecular race" in the levels of the corresponding master regulators (Kuchina *et al*, 2011). By placing them under independent optogenetic control, the means by which their dynamics impact these cellular decisions could be evaluated more easily and rigorously.

# Materials and Methods

## Bacterial strains

All systems utilize the *E. coli* BW29655 host strain (Zhou *et al*, 2003). The CcaSR system strain carries the pSR43.6 and pSR58.6 plasmids, which confer spectinomycin and chloramphenicol resistance, respectively (Schmidl *et al*, 2014). The Cph8-OmpR system strain carries the pSR33.4 (spectinomycin) and pSR59.4 (ampicillin) plasmids (Schmidl *et al*, 2014). The dual-system strain carries pSR58.6, pSR78 (spectinomycin), and pSR83 (ampicillin). Plasmid maps and sequences are available (Fig EV1 and Dataset EV1).

## Bacterial growth and light exposure

Cell culturing and harvesting protocols were developed to ensure a high degree of precision and reproducibility in experiments both from well-to-well and from day-to-day (Appendix Method S1). Cells were grown at 37°C and shaken at 250 rpm throughout the experiment (Sheldon Manufacturing Inc. SI9R) with temperature calibrated and logged by placing a thermometer probe in a sealed 125-ml water-filled flask (Traceable Excursion-Trac 6433). Cultures were grown in M9 media supplemented with 0.2% casamino acids, 0.4% glucose, and appropriate antibiotics. Precultures were prepared in advance by freezing 100-μl aliquots of early exponential phase cultures ($OD_{600} = 0.1–0.2$) grown in the same media conditions at −80°C (Appendix Method S2). Cultures were inoculated at low densities (typically $OD_{600} = 1 \times 10^{-5}$) to ensure that final densities did not reach stationary phase ($OD_{600} < 0.2$). For each experiment, 192 cultures were grown in 500 μl volumes within 24-well plates (ArcticWhite AWLS-303008), sealed with adhesive foil (VWR 60941-126).

Experiments were performed using eight 24-well LPA instruments (Gerhardt *et al*, 2016), enabling precise control of two LEDs to define the optical environment of 192 cultures at a time. LPA program files were generated using Iris (Gerhardt *et al*, 2016) and a custom Python tool (Dataset EV9).

## LED measurement

All LEDs were measured and calibrated (Appendix Method S3 and Dataset EV3) using a spectrometer (StellarNet UVN-SR-25 LT16) with NIST-traceable factory calibrations performed on both its

wavelength and intensity axes immediately prior to use for this study. A six-inch integrating sphere (StellarNet IS6) was used, enabling measurement of the total power output of each LED (in μmol s$^{-1}$). The spectrophotometer was blanked by a measurement of a dark sample before each LED measurement. Measurements were saved as .IRR files, which contain the complete LED spectral power density $P_{light}(\lambda)$ (μmol s$^{-1}$ nm$^{-1}$) in 0.5 nm increments as well as all setup parameters for the measurement (i.e., integration time and number of scans to average). These files were processed by Python scripts to calculate the LED characteristics, including the peak, centroid, FWHM, and total power. For spectral validation experiments, cinematic lighting filters (Roscolux) were cut, formed into LED-shaped caps, and fitted atop white LEDs (Table EV1).

## Calculation of $n_{light}$

Because the LEDs we utilize have fixed spectral characteristics, the spectral flux density (μmol m$^{-2}$ s$^{-1}$ nm$^{-1}$) incident on the photoreceptors can be parameterized by the LED intensity (μmol m$^{-2}$ s$^{-1}$). The cultures are shaken throughout the experiment, and we assume that the cells are well mixed within the culture volume. Thus, the mean light intensity within the culture volume, $n_{light}(\lambda)$, can be calculated by integrating the intensity throughout the volume of the well. Under the assumption of negligible light absorption by the culture sample (the M9 media is transparent, and the cultures are harvested at low density), this integral simplifies to become the total power of the LED (μmol s$^{-1}$) divided by the cross-sectional area of the well. Given a well radius of 7.5 mm, we calculate

$$n_{light}(\lambda) = \frac{P_{light}(\lambda)}{\pi(7.5 \times 10^{-3}\ \text{m})^2} \approx 5.659 \times 10^3\ \text{m}^{-2} \times P_{light}(\lambda)$$

## LED calibration

Each of the approximately 700 individual LEDs used in the study were measured (Appendix Method S3 and Dataset EV3), enabling compensation for variation in LED and LPA manufacturing (Tables EV1–EV3 and Dataset EV9). Each LED was calibrated while powered from the same LPA socket used in experiments. First, a sample of LEDs were measured to identify the electrical current required to achieve an appropriate level of total flux, $\int n_{light}(\lambda)\,d\lambda$. The amount of current required varied depending on the wavelength and manufacturer. The current was adjusted using the LPA "dot-correction (DC)" to achieve a total flux approximately 20% above 20 μmol m$^{-2}$ s$^{-1}$ when the LED was fully illuminated. The appropriate DC level was determined for each LED model. Using these DC levels, the complete set of LEDs were measured. LEDs that produced a total flux below 20 μmol m$^{-2}$ s$^{-1}$ were re-measured at a higher DC level. This set of LED measurements was used to convert the desired intensity time course of each LED into a series of 12-bit grayscale values (i.e., 0–4,095) used by the LPA. The LPA reads the grayscale values to produce the appropriate pulse-width-modulated (PWM) signal to achieve the desired intensities.

## Bacterial sample harvesting

Cultures were harvested for measurement (Appendix Method S1) after precisely 8-h growth by placing the 24-well plates into ice-water

baths. Each culture was then subjected to both an absorbance measurement to ensure consistent well-to-well and day-to-day growth, and flow cytometry for quantification of sfGFP or mCherry expression. Absorbance measurements were performed in black-walled, clear-bottomed 96-well plates (VWR 82050-748) in a plate reader (Tecan Infinite M200 Pro). Before fluorescence measurements were performed, culture samples were processed via a fluorescence maturation protocol to ensure measurements were representative of the total amount of produced fluorescent reporter (Olson *et al*, 2014). Rifampicin (Tokyo Chemical Industry R0079) was dissolved in phosphate-buffered saline (PBS, VWR 72060-035) at 500 µg/ml and used to inhibit sfGFP production during maturation.

### Flow cytometry

Population distributions of fluorescence were measured for each culture on a flow cytometer as previously described (Olson *et al*, 2014). A calibration bead sample (Spherotech RCP-30-5A) in PBS was measured immediately prior to the culture samples from each experimental trial. At least 5,000 events were collected for the calibration bead sample, and at least 20,000 events were collected for each culture sample.

### Flow cytometry data analysis

Single-cell distributions of sfGFP fluorescence were gated, analyzed, and calibrated into MEFL and MECY units using FlowCal (Castillo-Hair *et al*, 2016) via a custom Python script (Dataset EV9). Measurements were gated on the FSC and SSC channels using a gate fraction of 0.3 for calibration beads and 0.8 for cellular samples (Castillo-Hair *et al*, 2016). Reported culture fluorescence values are the arithmetic means of the cellular populations.

### Sensing model

The light-sensing model can be described by the following system of ODEs:

$$\frac{dS_g}{dt} = k_S + (k_2 + k_{dr}) \cdot S_a(t) - (k_1 + k_{dil}) \cdot S_g(t)$$

$$\frac{dS_a}{dt} = k_1 \cdot S_g(t) - (k_2 + k_{dil} + k_{dr}) \cdot S_a(t),$$

where the variables and rates have been previously introduced (Introduction, Results) with best-fit values summarized in Figs 2F and EV2F. Note that $k_1$ and $k_2$ are implicitly dependent upon time, as they are functions of the time-varying light environment of the sensors.

If we substitute for the fraction of active sensors, $y \equiv S_a / (S_g + S_a)$, the system can be expressed as:

$$\frac{dy}{dt} = k_1 - (k_1 + k_2 + k_{dil} + k_{dr}) \cdot y(t) = k_1 - k_{tot} \cdot y(t),$$

where

$$k_{tot} \equiv k_1 + k_2 + k_{dil} + k_{dr}.$$

This ODE can be solved analytically for a step change in light from one environment to another. If the step change occurs at time

$t = 0$, then $k_1$, $k_2$, and $k_{tot}$ are all fixed for $t > 0$. Given an initial sensor fraction $y(0) = y_0$, we find.

$$y(t) = y_0 + \left( \frac{k_1}{k_{tot}} - y_0 \right) \cdot \left( 1 - e^{-k_{tot}t} \right).$$

This solution represents an exponential transition from an initial sensor fraction of $y_0$ to a final fraction given by $k_1/k_{tot}$ with a time constant set by $k_{tot}$. As a result, we anticipate that the transition dynamics of $y(t)$ will be slowest under zero illumination when $k_{tot} = k_{dil} + k_{dr}$. We also expect that the transition rates will be unbounded as intensity increases.

Finally, for multiple light sources, we simply linearly combine the photoconversion rates from each source: $k_i = k_{i,\text{source 1}} + k_{i,\text{source 2}}$.

### TCS signaling model

We utilize a highly simplified model of TCS signaling and gene regulation. This model relates the production rate of the output gene $k_G(t)$ to the active ratio of light sensors $\frac{S_a(t)}{S_g(t)} = \frac{y(t)}{1-y(t)} \equiv R(t)$. We model TCS signaling as a pure time delay $\tau$ and a sigmoidal Hill function. For CcaSR, the Hill function is activated by increasing sensor ratios, while for Cph8-OmpR, the inverted TCS signaling activity results in a repressing Hill function. Thus, we write $k_G(t) = \hat{b} + \hat{a} \frac{R(t-\tau)^n}{K^n + R(t-\tau)^n}$ for CcaSR and $k_G(t) = \hat{b} + \hat{a} \frac{K^n}{K^n + R(t-\tau)^n}$ for Cph8-OmpR, where the variables and rates have been previously introduced (Introduction, Results) with best-fit values summarized in Figs 2F and EV2F.

### Output gene expression model

We model output gene expression by first-order production and dilution dynamics:

$$\frac{dG}{dt} = k_{G(t)} - k_{dil} \cdot G(t),$$

where the variables and rates have been previously introduced (Introduction, Results) with best-fit values summarized in Figs 2F and EV2F.

### Generation of model simulations

Simulations (Datasets EV2, EV7 and EV8) were produced by numerically integrating the system of ODEs using Python's scipy.integrate.ode method using the "zvode" integrator with a maximum of 3,000 steps (Dataset EV4).

### Model parameterization

The CcaSR and Cph8-OmpR models were parameterized using global fits of the model parameters to the complete training datasets (Figs 2B–E and EV2B–E, and Datasets EV2 and EV7). The "lmfit" Python package, which is based on the Levenberg-Marquardt minimization algorithm, was used to perform the fits and analyze the resulting parameter sets (Newville *et al*, 2014). The fits were performed by minimizing the sum of the square of the relative error between each measured data point and the same point in a corresponding model simulation. Thus, the form of the error metric

utilized was error $= \sum_i \left( \frac{G_i^{\text{data}} - G_i^{\text{model}}}{G_i^{\text{data}}} \right)^2$ across the complete set of data points $\{ G_i^{\text{data}} \}$.

### Estimation of PCSs

Photoconversion cross section estimates $\sigma_i^{\text{est.}}(\lambda)$ were constructed by linearly regressing a cubic spline to the experimentally determined photoconversion rates in order to produce a continuous PCS (Appendix Fig S5 and Dataset EV5). The $\sigma_i^{\text{est.}}(\lambda)$ were produced by minimizing the error between unit experimental photoconversion rates $\hat{k}_i^{\text{expt.}}$ (Figs 2F and EV2F) and spline-derived predictions $\hat{k}_i^{\text{pred.}} = \int \sigma_i^{\text{est.}} \cdot \hat{n}_{\text{light}} \, d\lambda$. The splines were constructed by establishing a series of integral constraints for the photoconversion rates, continuity constraints for the spline knots, and boundary constraints. As this problem contains more constraints than parameters, optimization is required. We used weighted least-squares with Lagrange multipliers to optimize each spline. To avoid over-parameterization of the $\sigma_i^{\text{est.}}(\lambda)$, we used "Leave-one-out cross-validation (LOOCV)" to evaluate the performance of splines with between 5 and 20 knots to determine the ideal number required for each The $\sigma_i^{\text{est.}}(\lambda)$ (Appendix Fig S6). The resulting optimal number of splines was 12 and 8 for CcaS $\sigma_g$ and $\sigma_a$ and 12 and 12 for Cph8 $\sigma_g$ and $\sigma_a$. One knot was fixed at 1,050 nm, and the remaining knots were evenly distributed between 350 and 800 nm (Appendix Figs S5 and S6).

### Calculation of prediction error (RMSEs)

For model validation, we use a relative error metric (RMSE $= \sqrt{ \left( \sum_i \log_{10}(G_i^{(\text{pred.})}/G_i^{(\text{expt.})}) \right)^2 / n }$) that reports the root-mean-square (RMS) of the $\log_{10}$ error between the predicted and measured responses (Datasets EV2, EV7 and EV8).

### Light program generator (LPG) algorithm

The LPG was used as previously described (Olson *et al*, 2014). The only modification was to use simulations generated by the model described herein rather than the previous model. Compensated light programs were generated by incorporating the presence of the external light signal into the model simulations.

### Comparison of output gene expression ranges for single- vs. dual-systems

The CcaSR output range is nearly conserved (60-fold vs. 56-fold), while the mCherry response from Cph8-OmpR is substantially reduced (210-fold vs. 6.0-fold). Additionally, the light response is less sensitive than was observed for Cph8-OmpR individually, as half-repression requires a 5.2-fold higher intensity (Appendix Fig S14). We speculate that the reduction in the output range and decrease in sensitivity of Cph8-OmpR results from a competition between Cph8 and CcaS for limiting PCB, leading to a substantial population of light-insensitive apo-Cph8. Notably, the growth rate (Appendix Fig S13) of the dual-system strain (39.2 min per doubling) is only marginally slower than the single-system strains (37.4 and 37.9 min for CcaSR and Cph8-OmpR, respectively).

### Detailed descriptions of multiplexed function generation reference signals

In the below descriptions of the multiplexed function generation reference signals (Dataset EV8), the percentages and fractions correspond to a log-scaled representation of the output range (e.g., if a system has a 16-fold output range, the 50% level on a log scale would be at the same expression at the 25% level on a linear scale).

1. Dual-sines. The mCherry reference signal is described by the function $0.5 + 0.3 \sin(2\pi t / 480 \, \text{min})$ while the sfGFP reference signal follows $0.7 + 0.2 \sin(2\pi t / 360 \, \text{min})$.
2. Sine and steps. The mCherry reference signal is the same as in "Dual-sines", while the sfGFP signal is a series of 80-min holds and 40-min linear ramps in increasing increments of 20% of the output range.
3. The sfGFP signal is the same as in the "Sine and steps" program, while the mCherry signal is the inverse of the same program.
4. (Time-shifted waveform). The mCherry signal is a complex function consisting of the following steps:

   a. linear ramp from 0 to 70% over 80 min,
   b. hold at 70% for 40 min,
   c. linear ramp down to 50% over 60 min,
   d. hold at 50% for 40 min,
   e. sinusoidal signal described by the function $0.5 + 0.25 \sin(2\pi(t - 220 \, \text{min})/220 \, \text{min})$,
   f. hold at 50% for 40 min.

The sfGFP signal is the same program but delayed by 60 min.

**Expanded View** for this article is available online.

### Acknowledgements

We thank Sebastian Schmidl for providing the dual-system strain, Prabha Ramakrishnan and Karl Gerhardt for helpful discussions on the design of the characterization experiments, Sebastian Castillo-Hair for helpful ideas on constructing the photoconversion cross-sectional estimates, Lucas Hartsough for providing an LPA programming script, and Keshav Rao for assistance with data collection. We would also like to thank Dr. Joel Moake and his lab for use of the flow cytometer and plate reader. This work was supported by the Office of Naval Research (MURI N000141310074) and an NSF CAREER award (1553317).

### Author contributions

EJO and JJT conceived the project. EJO designed and performed experiments and analyzed data. CNT assisted with construction of the photoconversion cross-section estimates and design/performance of spectral validation trials. EJO and JJT wrote the manuscript.

### Conflict of interest

The authors declare that they have no conflict of interest.

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
