## [Review Process File · Molecular Systems Biology]

A photoconversion model for full spectral programming and multiplexing of optogenetic systems

Evan Olson, Mr. Constantine Tzouanas and Jeffrey Tabor

Corresponding author: Jeffrey Tabor, Rice University

Review timeline:

Submission date:	21 November 2016
Editorial Decision:	10 January 2017
Revision received:	22 February 2017
Accepted:	22 March 2017

Editor: Maria Polychronidou

Transaction Report:

1st Editorial Decision

10 January 2017

Thank you again for submitting your work to Molecular Systems Biology. We have now heard back from the three referees who agreed to evaluate your study. As you will see below, they all appreciate that the study is interesting and is going to be useful for the field. However, they list several concerns, which we would ask you to address in a revision.

Without repeating all the points listed below, since they are rather clear, reviewers #1 and #3 point out that including further analyses demonstrating the applications of the framework would significantly enhance the impact of the study. The two reviewers provide constructive suggestions in this regard. Please let me know in case you would like to further discuss any of the issues raised by the referees.

Reviewer #1:

Summary

In this work, Olsen et al expanded upon the previously published photoreversible two component systems (TCS) CcaSE and Cph8-OmpR. In particular, the authors aimed to generalize their previous model by parameterizing it so that it could accurately predict the dynamic response to any combination of input light signals. The authors demonstrate that the model can indeed be generalized to multiple different types (and mixtures) of light signals with only minimal calibration, and demonstrates the functional utility of it by programming distinct non-overlapping expression control of both two component systems integrated into the same cell.

Review

The study presents a new approach that can potentially standardize the measurements and calibration of optogenetic tools, which represents the major conceptual advance of the work. In

particular, previous models have to be calibrated from device to device. Using the new modeling framework, a research can calibrate their models based on several fundamental parameters (that're are less dependent on the specific instruments). The authors provide a fine proof of concept using their previously published data sets. That said, I believe this conceptual advance should be better clarified and emphasized.

Moreover, I found much of the calibration experiments to be drawn out and unnecessary to include in the main text. Instead, I found the multiplexed biological function generation portion of this paper (Fig. 7) to be the most significant aspects of the work and therefore should be elaborated. I have two suggestions that would have satisfactorily differentiated this paper from the previous ones: First, the authors could expand upon the functionality of dual light systems, and integrate it with systems that have been previously shown to suffer from non-orthogonal or leaky expression interference. Second, the authors could significantly expand upon the notion that any light source could be used with this model. Instead of using two different systems with relatively distant activating wavelengths, it would have been useful to see directed mutagenesis of only one TCS (e.g. either CcaSR OR Cph8-ompR), to demonstrate that mutants with slightly different activating wavelengths can be used in tandem with the same degree of orthogonality, therefore limiting the amount of orthogonal light-inducible systems only by what is possible to generate through mutation (e.g. not very limited).

Other comments:

1. Cubic-spline fitting is rather complex and sometimes arbitrary, requiring unknown functions with multiple parameters - I am wondering whether there is a potential for over-fitting, and the claims that this works with all different light sources might be overstated? This is an important question as it ties back to the central conceptual advance of the work. Alternatively, is there a simpler function that will have a lesser fit but more generalizable, and how much predictive accuracy do you lose when downgrading the fit here?

2. Why were bacterial cultures pre-frozen at -80 in aliquots? Does that mean that all experiments were actually using the exact same clone, and if you switch even a clone of the same strain the prediction would be inferior to what is shown? How many different cultures was this tested with? Can you demonstrate the dual functionality in a different strain to demonstrate generality?

3. How long can the experiments run for? Would the control survive multi-day seasonal dilutions or do mutants overtake fast - especially when exposed to two different light sources, is there an increased rate at which mutants can expect to arise and how does this influence the experiments?

4. Figure 4 needs to be clarified in the legend: what is predictive? what is the input? And how was it generated?

Reviewer #2:

Summary and general remarks:

In this work, Olson et al. develop a parameterized model to predict the output of (their) two previously engineered transcriptional circuits in *E. coli* based on two-component signaling systems, in which reporter gene expression is under the dynamic (optogenetic) regulation by one or multiple wavelengths of light. Through highly detailed characterizations of the engineered bacterial strains and also of their opto-electronic hardware, they construct a model that predicts the response under far more spectrally complex demands than their previous work. The experiments are exceptionally well executed, and the results are convincing,

While it is unclear whether the model represents a major conceptual advance as one that combines existing models of TCS and photoconversion in bi-stable photoreceptors, the tremendous strengths of this paper are in the technical advance, namely in the prediction capabilities in response to highly spectrally complex and dynamic stimuli. The overall dataset is impressive in its rigor and completeness. Few if any synthetic biological and/or optogenetic systems have been characterized to this detail or driven to this level of reliability.

Based on the technical advance and convincing results, I recommend this manuscript for publication, and I believe that it will be of interest to the journal's readership.

Major points:
None

Minor points

- The use of a cubic spline in the CCAS action spectrum in Figure 3a gives rise to a shape that is very different to the absorbance spectrum of this particular photoreceptor (Hirose et al. PNAS 2008), and would suggest a third stable state to account for. The appropriateness of the fit should be discussed in light of the fact that a photoreceptor absorbance spectrum is not necessarily a circuit's functional action spectrum.

- Likewise, the explanation of the 526nm data point as an outlier in Figure 3a and S6 should be further addressed. If the origin of this deviation is that LED has a dramatically larger solid angle than the others, it is testable using other dome-LEDs with other solid angles, via the use of a GRIN lens to focus, or corrected for based on photometer measurements through an iris the diameter of the plate well. It is possible that the origin is actually spectral in nature as opposed to photon fluence, given that the green-minus-red difference spectrum of the photoreceptor dips around this wavelength (Hirose et al. PNAS, 2008)

- The manuscript and model would be much easier to understand if there were equation numbers, or at least if variables were always defined the variables in the sentence they are introduced.

Reviewer #3:

Comments to Authors:

Summary: In this article, the authors report models of the CcaSR and Cph8-OmpR two-component optogenetic systems. These models are an extension of a two-state model of photoreceptor activation that has been used in vitro (Butler, et al 1964) as well as an improvement on the authors' previous phenomenological model (Olson, et al 2013). These models allow the authors to design optical inputs to control gene expression (of fluorescent proteins) downstream of the CcaSR and Cph8 systems. Most impressively, the authors demonstrate that they can simultaneously drive specific gene expression programs from both optogenetic systems in the same cell, that is, they are able to account for cross-stimulation such that both optogenetic systems give the desired gene expression output despite conflicts in the illumination for one optogenetic system relative to the other. This ability to drive appropriate gene expression output from an optogenetic system using their parameterized models, despite additional illumination programs, should also give researchers the ability to utilize optogenetic systems in combination with fluorescent reporters which require different wavelengths for excitation. This in my opinion is a particularly powerful potential application of the authors' work.

The great promise of optogenetics in quantitative cell biology is the ability to use an optogenetic system to drive the activity and/or level of particular proteins and understand how this perturbation propagates through cellular networks to regulate other cellular networks and cellular physiology/behavior, what timescales are important, and even how one might design a perturbation to get optimal cellular outcome (for example, optimal output from a synthetic biological network). While the number of optogenetic systems available has proliferated in the last ten years, the amount of characterization of these systems needed in order to have such precise control has lagged far behind. This study, particularly in conjunction with the authors' previous work designing open source characterization software/hardware and quantification methods (Gerhardt, et al 2016, Castillo-Hair, et al 2016) is an important first-step in unlocking this important potential of optogenetic perturbation techniques.

In particular, I believe that this is the first study to demonstrate the possibility of simultaneously

controlling two optogenetic systems (to generate very impressive gene expression patterns I might add) and the potential to utilize modeling to design light inputs such that one can compensate for illumination required to utilize fluorescent reporters. The ability to multiplex optogenetic systems and compensate for fluorescence measurements will be very important for the field moving forward. Overall the paper is well written. The supplemental materials are excellent and will serve immediately as an important resource for the optogenetic/synthetic biology/quantitative cell biology communities. I applaud the authors' efforts (evident from the supplemental material here as well as the Gerhard, et al and Castillo-Hair, et al publications) to create open-source optogenetic characterization resources that should help this technology to be more widely adopted.

If I have one criticism of the work, it is that it doesn't go far enough. A lot of lip-service has been given to using real-time control techniques to drive interesting inputs into biological networks, however, in most cases only patterns of fluorescent proteins are generated (Melendez, et al 2014, Miliadis-Argeitis, et al 2011, etc). In my opinion, this paper would be more compelling if the authors' had done the additional step(s) of (1) driving two different transcription factors, enzymes, or synthetic pathway components under the control of the optogenetic systems and showing that their model was still able to provide control and/or (2) driving activity of a transcription factor or enzyme and assaying its activity using a fluorescent reporter, thus demonstrating the compensation control when also doing fluorescence protein excitation. I am tentative to voice this criticism, as it isn't really a criticism per se. More an expression of what it would be nice to see in an idea world. The research in the publication as-is represents a significant amount of high-quality work and is suitable for publication.

Some specific questions and concerns are outlined below:

Questions and Comments:

- In Figure 3, can you elaborate on why the prediction for LED 6 is not as good as for the other LEDs? Where is the breakdown happening in your model? Is the expectation that your model will perform poorly for all LEDs that drive intermediate expression levels?
- In Figure 4a and 4b it looks like the light input for the UV LED ("UV Mono") and the 526nm LED ("Green Mono") are identical. Is this actually the case, and if so could an explanation be given in the text for why one would expect this to be the case? Particularly given the relative poorness of the performance of the UV Mono input to give the desired output. Also, related to this figure you state that no significant toxicity in terms of cell growth was seen for the UV light exposure. Since one might naively expect the shorter wavelengths to cause increased toxicity this data should be shown, as it isn't clear what "no significant toxicity" means. Does that mean nothing obvious was seen while doing the light-exposure experiments or was this actually carefully measured?
- Figure 6. This is a beautiful illustration of the ability of your model to allow multiplexing of optogenetic systems. However, as I stated in the initial summary, the disconnect between this figure and doing interesting experiments using optogenetics is that there is a big difference between driving relatively inert GFP/mCherry and a protein that has physiological effects on the cell. How would the model need to change if you were driving a protein that for instance, slowed cell growth or affected metabolism?
- The Python scripts used for parsing input files and fitting models should be provided as supplementary material. Additionally, since in my mind the biggest strength of this article is that it could be repeated for other optogenetic systems (or at least, other researchers could use this article as a guide) these scripts should be well-documented, perhaps with illustrative examples.
- In the Dynamic and spectral characterization of CcaSR results section, there is no discussion of why the four characterization experiments were picked. Are these optimal for some reason or were they simply found to be sufficient to get a predictive model? Are they expected to work for most optogenetic systems for which one might try to repeat the authors' methodology?
- In general it is difficult at times to follow what the different pieces of the model are. Sometimes they are called by different module names (i.e. "TCS signaling model") and other times as just "the Hill function". Also, the different k's get quite confusing, particularly when the k from the Hill function is mentioned.

Response to reviewer comments

Reviewer #1:

Summary

In this work, Olsen et al expanded upon the previously published photoreversible two component systems (TCS) CcaSE and Cph8-OmpR. In particular, the authors aimed to generalize their previous model by parameterizing it so that it could accurately predict the dynamic response to any combination of input light signals. The authors demonstrate that the model can indeed be generalized to multiple different types (and mixtures) of light signals with only minimal calibration, and demonstrates the functional utility of it by programming distinct non-overlapping expression control of both two component systems integrated into the same cell.

Review

The study presents a new approach that can potentially standardize the measurements and calibration of optogenetic tools, which represents the major conceptual advance of the work. In particular, previous models have to be calibrated from device to device. Using the new modeling framework, a research can calibrate their models based on several fundamental parameters (that're are less dependent on the specific instruments). The authors provide a fine proof of concept using their previously published data sets. That said, I believe this conceptual advance should be better clarified and emphasized.

We thank the Reviewer for this suggestion. We have updated the final paragraph of the introduction to clarify the conceptual and technical advances of this work: the optogenetic TCS model and the detailed characterization experiments. Additionally, we have added text to distinguish the conceptual advance from the advantages and novel capabilities that it provides (including the standardization of measurement of optogenetic systems and device-to-device reproducibility) in the Discussion.

We would also like to clarify that the data sets included in this manuscript have not been previously published.

Moreover, I found much of the calibration experiments to be drawn out and unnecessary to include in the main text. Instead, I found the multiplexed biological function generation portion of this paper (Fig. 7) to be the most significant aspects of the work and therefore should be elaborated.

We appreciate the Reviewer's concern. We agree that the demonstration of multiplexed function generation is one of the most significant and exciting aspects of this work. However, we believe that our presentation of the characterization experiments is also of great interest to many among our target audience. The unprecedented resolution of both our temporal and spectral datasets provides an exceptionally clear window into the signal transduction properties of CcaSR and Cph8-OmpR, and the technical advances we have made in order to execute these experiments are fundamental to the performance of our model.

We believe that our presentation of these results in the main text is already brief (300 words and one figure between two paragraphs in the results). We only show the CcaSR data in the main text. The Cph8-OmpR data is shown in the Appendix (Figure S10). In contrast, the development, presentation, and discussion of multiplexed function generation constitutes over 800 words between the results and discussion and two figures (Fig. 6,7).

I have two suggestions that would have satisfactorily differentiated this paper from the previous ones: First, the authors could expand upon the functionality of dual light systems, and integrate it with systems that have been previously shown to suffer from non-orthogonal or leaky expression interference.

This is an excellent suggestion. We strongly agree that one of the clearest demonstrations of our approach is that we can independently program the outputs from two non-orthogonal light sensing TCSs in the same cell. We believe that our choice of the CcaSR and Cph8-OmpR demonstrates this capability very well, as these systems exhibit major spectral cross-reactivity. While the activating wavelengths of these two systems are well-separated (green versus far red), the deactivating wavelengths of CcaS strongly overlap with the activating wavelengths of Cph8 (both red). Indeed, in the manuscript, we illustrate that red light used to control Cph8-OmpR has a major unwanted perturbative effect on CcaSR (Fig. 4c). We go on to show this cross-talk in the dual light sensor system in Fig. 5b.

Second, the authors could significantly expand upon the notion that any light source could be used with this model. Instead of using two different systems with relatively distant activating wavelengths, it would have been useful to see directed mutagenesis of only one TCS (e.g. either CcaSR OR Cph8-ompR), to demonstrate that mutants with slightly different activating wavelengths can be used in tandem with the same degree of orthogonality, therefore limiting the amount of orthogonal light-inducible systems only by what is possible to generate through mutation (e.g. not very limited).

We again argue that CcaS and Cph8 represent a non-orthogonal pair of light sensors, as clarified above. We are certainly intrigued by this second suggestion, as we are also curious to examine the full capabilities of our approach. However, we feel that such a demonstration is significantly beyond the scope of this study. Directed evolution of cyanobacteriochrome or phytochrome light sensors to tune their action spectra to desired wavelengths while retaining proper photoswitching is a very challenging prospect. On the other hand, we do envision a follow-up manuscript focused on the development of a higher order multiplexer incorporating three or more light systems within the visible spectrum, as development of such a system would clearly motivate the extent to which highly spectrally-coupled systems can be independently regulated.

Other comments:

1. Cubic-spline fitting is rather complex and sometimes arbitrary, requiring unknown functions with multiple parameters - I am wondering whether there is a potential for over-fitting, and the

claims that this works with all different light sources might be overstated? This is an important question as it ties back to the central conceptual advance of the work. Alternatively, is there a simpler function that will have a lesser fit but more generalizable, and how much predictive accuracy do you lose when downgrading the fit here?

We agree with the Reviewer's assessment that cubic-spline fitting can easily lead to overfitting, resulting in poor predictive performance for data beyond the training data set. Indeed, this problem motivated our use of a cross-validation approach in which the training data set is partitioned into both training and testing data sets in order to better evaluate the generalizability of the spline. We used this cross-validation approach to guide our selection of the complexity (number of knots) used for both the activating and deactivating PCSs for each system. We direct the reviewer to our explanation our approach detailed in the "Estimation of PCSs" section of the Materials and Methods, and to Supplementary Fig. 7 in which we show the cross-validation results.

2. Why were bacterial cultures pre-frozen at -80 in aliquots? Does that mean that all experiments were actually using the exact same clone, and if you switch even a clone of the same strain the prediction would be inferior to what is shown? How many different cultures was this tested with? Can you demonstrate the dual functionality in a different strain to demonstrate generality?

These are excellent questions. The use of clonal -80 °C preculture aliquots has long been used in microbiology and related fields to reduce day-to-day variability in experimental results arising from subtle physiological differences in cultures. We opted to utilize this method to better isolate the desired observations (i.e. light-driven signal transduction) from extrinsic, experimental factors such as clonal, day-to-day, and population variation. While certainly intriguing, we believe that analyses of the impact of extrinsic experimental factors upon our experimental systems and models deserve their own study and are beyond the focus of this work.

3. How long can the experiments run for? Would the control survive multi-day seasonal dilutions or do mutants overtake fast - especially when exposed to two different light sources, is there an increased rate at which mutants can expect to arise and how does this influence the experiments?

This is a fundamental unsolved problem in synthetic biology. Mutants can always arise in every synthetic biological system engineered to date. The engineering of systems that are robust to mutations would be of tremendous general interest and is outside the scope of this work.

4. Figure 4 needs to be clarified in the legend: what is predictive? what is the input? And how was it generated?

We thank the reviewer for bringing this to our attention. We have clarified the text of the caption.

Reviewer #2:

Summary and general remarks:

In this work, Olson et al. develop a parameterized model to predict the output of (their) two previously engineered transcriptional circuits in E. coli based on two-component signaling systems, in which reporter gene expression is under the dynamic (optogenetic) regulation by one or multiple wavelengths of light. Through highly detailed characterizations of the engineered bacterial strains and also of their opto-electronic hardware, they construct a model that predicts the response under far more spectrally complex demands than their previous work. The experiments are exceptionally well executed, and the results are convincing,

While it is unclear whether the model represents a major conceptual advance as one that combines existing models of TCS and photoconversion in bi-stable photoreceptors, the tremendous strengths of this paper are in the technical advance, namely in the prediction capabilities in response to highly spectrally complex and dynamic stimuli. The overall dataset is impressive in its rigor and completeness. Few if any synthetic biological and/or optogenetic systems have been characterized to this detail or driven to this level of reliability.

Based on the technical advance and convincing results, I recommend this manuscript for publication, and I believe that it will be of interest to the journal's readership.

We sincerely appreciate these comments, and share the Reviewer's thoughts.

Major points:

None

Minor points

- The use of a cubic spline in the CCAS action spectrum in Figure 3a gives rise to a shape that is very different to the absorbance spectrum of this particular photoreceptor (Hirose et al. PNAS 2008), and would suggest a third stable state to account for. The appropriateness of the fit should be discussed in light of the fact that a photoreceptor absorbance spectrum is not necessarily a circuit's functional action spectrum.

We agree that it is worthwhile to include a statement regarding the inadequacy of the absorbance spectrum alone for capturing the photoreceptors behavior. We have included a sentence in the 2nd-to-last paragraph of the introduction to make this clear.

- Likewise, the explanation of the 526nm data point as an outlier in Figure 3a and S6 should be further addressed. If the origin of this deviation is that LED has a dramatically larger solid angle than the others, it is testable using other dome-LEDs with other solid angles, via the use of a GRIN lens to focus, or corrected for based on photometer measurements through an iris the diameter of the plate well. It is possible that the origin is actually spectral in nature as opposed to photon fluence, given that the green-minus-red difference spectrum of the photoreceptor dips around this wavelength (Hirose et al. PNAS, 2008)

We thank the reviewer for these useful suggestions. We have now investigated this LED further and identified that it is not substantially clipping due to having a wide solid angle, as we initially thought. We have come to this conclusion by performing two additional measurements: 1) analyzing a photograph of the LED with a diffuser material at the plane of the aperture, and 2) measuring the gene expression response of CcaSR to a set of LEDs with identical spectral characteristics and different molded lenses (producing a range of solid angles). After calibrating these LEDs using the integrating sphere (as described in our supplemental Detailed LED Measurement Protocol), we observed no change in gene expression up to the 30 degree half-angle present in the 526 nm LED in question.

Given these new results, we are left with the conclusion that the observed reduction in the response of CcaSR to this LED is due to the spectral characteristics of CcaS, as the Reviewer suggests. We have therefore re-constructed our photoconversion cross-section estimates for both the CcaSR and Cph8-OmpR systems without excluding this LED, and updated the main text, figure (**Fig. 3**), and Appendix figures (**Fig. S6,7,14**) accordingly. We find that this change resulted in a minor increase in the RMSEs for the spectral validation experiments.

- The manuscript and model would be much easier to understand if there were equation numbers, or at least if variables were always defined the variables in the sentence they are introduced.

We apologize for this oversight. We have updated the main text in the Introduction and in the first paragraph of the results to ensure that a description is provided in each instance a variable is introduced. We have also included additional cross-references to the parameter tables in Fig. 2f and Fig. S8f in the model sections of the Materials and Methods. Finally, we have replaced the Hill parameter ‘k’ with ‘K’ in order to more readily distinguish it from the various subscripted rate parameters. We believe these edits substantially improve the presentation and clarity of the manuscript, model, and variables.

Reviewer #3:

Comments to Authors:

Summary: In this article, the authors report models of the CcaSR and Cph8-OmpR two-component optogenetic systems. These models are an extension of a two-state model of

photoreceptor activation that has been used in vitro (Butler, et al 1964) as well as an improvement on the authors' previous phenomenological model (Olson, et al 2013). These models allow the authors to design optical inputs to control gene expression (of fluorescent proteins) downstream of the CcaSR and Cph8 systems. Most impressively, the authors demonstrate that they can simultaneously drive specific gene expression programs from both optogenetic systems in the same cell, that is, they are able to account for cross-stimulation such that both optogenetic systems give the desired gene expression output despite conflicts in the illumination for one optogenetic system relative to the other. This ability to drive appropriate gene expression output from an optogenetic system using their parameterized models, despite additional illumination programs, should also give researchers the ability to utilize optogenetic systems in combination with fluorescent reporters which require different wavelengths for excitation. This in my opinion is a particularly powerful potential application of the authors' work.

The great promise of optogenetics in quantitative cell biology is the ability to use an optogenetic system to drive the activity and/or level of particular proteins and understand how this perturbation propagates through cellular networks to regulate other cellular networks and cellular physiology/behavior, what timescales are important, and even how one might design a perturbation to get optimal cellular outcome (for example, optimal output from a synthetic biological network). While the number of optogenetic systems available has proliferated in the last ten years, the amount of characterization of these systems needed in order to have such precise control has lagged far behind. This study, particularly in conjunction with the authors' previous work designing open source characterization software/hardware and quantification methods (Gerhardt, et al 2016, Castillo-Hair, et al 2016) is an important first-step in unlocking this important potential of optogenetic perturbation techniques.

In particular, I believe that this is the first study to demonstrate the possibility of simultaneously controlling two optogenetic systems (to generate very impressive gene expression patterns I might add) and the potential to utilize modeling to design light inputs such that one can compensate for illumination required to utilize fluorescent reporters. The ability to multiplex optogenetic systems and compensate for fluorescence measurements will be very important for the field moving forward.

Overall the paper is well written. The supplemental materials are excellent and will serve immediately as an important resource for the optogenetic/synthetic biology/quantitative cell biology communities. I applaud the authors' efforts (evident from the supplemental material here as well as the Gerhard, et al and Castillo-Hair, et al publications) to create open-source optogenetic characterization resources that should help this technology to be more widely adopted.

If I have one criticism of the work, it is that it doesn't go far enough. A lot of lip-service has been given to using real-time control techniques to drive interesting inputs into biological networks, however, in most cases only patterns of fluorescent proteins are generated (Melendez, et al 2014, Miliias-Argeitis, et al 2011, etc). In my opinion, this paper would be more compelling if the authors' had done the additional step(s) of (1) driving two different transcription factors, enzymes, or synthetic pathway components under the control of the optogenetic systems and

showing that their model was still able to provide control and/or (2) driving activity of a transcription factor or enzyme and assaying its activity using a fluorescent reporter, thus demonstrating the compensation control when also doing fluorescence protein excitation. I am tentative to voice this criticism, as it isn't really a criticism per se. More an expression of what it would be nice to see in an idea world. The research in the publication as-is represents a significant amount of high-quality work and is suitable for publication.

We thank the Reviewer for their thoughtful consideration of our work and its impact. We certainly agree that a demonstration that our dual-system function-generation approach can be used to control genes other than fluorescent proteins (e.g. transcription factors or enzymes) would make our demonstrations more compelling. However, we argue that our previous study with the TetR repressor and its target promoter $P_{\text{LtetO-1}}$ (Olson et. al. Nature Methods 2014) provides a sufficient proof-of-concept that we can use our light sensors to program the expression dynamics of transcription factors in order to study the dynamics of gene regulatory circuits. That being said, we share the Reviewer's interest in using the optogenetic multiplexing approach we report in this manuscript to perform new types of interrogations of a variety of biological systems (as we highlight in the Discussion). However, we ultimately believe that such studies are better suited to a follow-up manuscript, particularly given the already technically-dense nature of the current manuscript.

Some specific questions and concerns are outlined below:

Questions and Comments:

- In Figure 3, can you elaborate on why the prediction for LED 6 is not as good as for the other LEDs? Where is the breakdown happening in your model? Is the expectation that your model will perform poorly for all LEDs that drive intermediate expression levels?

The Reviewer poses an excellent question. We do indeed believe that poor performance of the prediction of LED 6 relative to the other LEDs results from the intermediate response of CcaSR to this source, even at high light intensities.

We can coarsely categorize the response of the system to light sources into three scenarios: strongly activating, strongly deactivating, and intermediate. In the case of strongly activating and deactivating sources, we observe that at high intensities, the system will be either fully activated or fully deactivated. This gene expression level of these high-intensity responses is independent of the exact values of the forward and reverse photoconversion rates. For example, compare the high-intensity responses of CcaSR to LEDs 1-5, which are all strongly-activating sources. Despite the substantial variation in the predicted photoconversion rates for these LEDs, the high-intensity response does not change. Instead, the difference in the model prediction for these LEDs is captured in the intensity of light required to achieve activation (i.e. the sensitivity to the photoconversion rates is primarily captured by translations along the x-axis).

However, when the light sources drive an intermediate response, changes in the photoconversion rates directly affect the gene expression levels at high light intensities. In this case, we observe that the sensitivity to the photoconversion rates is captured by the asymptotic behavior of the predicted curves at high intensities. While the sensitivity observed in the x-axis shifts is present for all three scenarios, the sensitivity in the asymptotic behavior is only present for intermediate responses. Thus, given that the total sensitivity to errors in photoconversion rates is the combination of these two sources, we argue that this represents the mechanism underlying the increased error observed in light sources producing intermediate responses.

- In Figure 4a and 4b it looks like the light input for the UV LED ("UV Mono") and the 526nm LED ("Green Mono") are identical. Is this actually the case, and if so could an explanation be given in the text for why one would expect this to be the case? Particularly given the relative poor performance of the UV Mono input to give the desired output.

We thank the Reviewer for their careful attention to detail and for bringing notice to this potential issue. We have investigated our records of the light programs used for these experiments and confirmed that the correct light programs were used. We recognize the potential for confusion, given the similarity between the generated programs. We have constructed a graphic to compare the two programs (below) demonstrating that despite their apparent similarity, there are indeed substantial differences between them. Most notably, the Green LED program maintains a ~25% higher light intensity than the UV program at throughout the majority of the program.

Also, related to this figure you state that no significant toxicity in terms of cell growth was seen for the UV light exposure. Since one might naively expect the shorter wavelengths to cause increased toxicity this data should be shown, as it isn't clear what "no significant toxicity" means. Does that mean nothing obvious was seen while doing the light-exposure experiments or was this actually carefully measured?

We measured the initial and final culture densities for each sample in this manuscript, enabling us to investigate the potentially phototoxic effects of the various light sources. We wrote that the “Mono UV” experiment exhibited no significant toxicity, as the final OD measurements for this experiment were statistically identical to the final OD measurements for the other three dynamic validation experiments in Figure 4 (see figure below).

However, we agree with the reviewer’s suspicion that LEDs in the shorter wavelength range could cause growth inhibition. Using the forward/reverse action spectrum measurements (Fig 2c,d) we have investigated whether these cultures, which were exposed to LEDs of various intensities and wavelength ranges for a period of eight hours, exhibited any growth inhibition. We indeed found that at the shortest wavelengths, at the highest intensities ($>5 \mu\text{mol m}^{-2} \text{s}^{-1}$) there is a marked reduction in the final cell density (see figure below, circles are forward-spectrum measurements (Fig 2c) and square are reverse-spectrum measurements (Fig 2d)).

We suspect that the lack of any observable growth defects in the “Mono UV” experiment is a result of the very short duration in which the cells are exposed to growth-inhibiting intensities (10 minutes). We maintain our original suspicion, stated in the main text, that the error in the model prediction for this experiment is a result of photodegradation of the phycocyanobilin (PCB) chromophore.

We have added Appendix figures (Figures S8,9) detailing these observations.

- Figure 6. This is a beautiful illustration of the ability of your model to allow multiplexing of optogenetic systems. However, as I stated in the initial summary, the disconnect between this figure and doing interesting experiments using optogenetics is that there is a big difference between driving relatively inert GFP/mCherry and a protein that has physiological effects on the cell. How would the model need to change if you were driving a protein that for instance, slowed cell growth or affected metabolism?

The reviewer brings up an excellent point regarding the ability to control gene expression when the expression of the gene itself contributes to physiological changes in the cell. In order to maintain use of a predictive model alone to control expression signals, model changes would be required to account for changes. We suspect it may be possible to account for these changes via the growth rate parameter, as we have shown in our previous work (Olson et al. Nature Methods, 2014). However, it is possible that gene toxicity could produce effects beyond the growth rate parameter (e.g. transcription or translation rates), and in this case additional experiments would

be required to parameterize a model of the effects of the toxicity. Such models could actually provide insight into more general principles of how to manage the unintended interactions present in many engineered biological systems that occur due to cellular resource sharing. We find this suggestion to be an excellent starting point for a follow-up work. In addition, it bears mentioning that our current model could be complemented by real-time feedback approaches that other groups have pioneered (as mentioned in the Discussion) in order to compensate for unintended or unforeseeable perturbations to the optogenetic signaling behavior.

- The Python scripts used for parsing input files and fitting models should be provided as supplementary material. Additionally, since in my mind the biggest strength of this article is that it could be repeated for other optogenetic systems (or at least, other researchers could use this article as a guide) these scripts should be well-documented, perhaps with illustrative examples.

We agree and hope that our approach can serve as a guide for the characterization and analysis of other optogenetic systems. We have substantially reworked the supplemental datasets to include all data and code used in this manuscript, including: a) experimental data (Dataset EV2,7,8), b) all model fitting, analysis, and plotting scripts (Dataset EV2,7,8), c) the LED calibration data and processing script (Dataset EV3), d) the model simulation code (Dataset EV4), e) the PCS spline estimation code (Dataset EV5), and f) the custom script we developed to automate generation of light programs and experimental data analysis (lpa-tools, Dataset EV9). We consider lpa-tools to be a target for continuous improvements (which we intend release as updates via a Github repository), as it was initially developed for personal use and requires much work to improve its user interface and feature set.

- In the Dynamic and spectral characterization of CcaSR results section, there is no discussion of why the four characterization experiments were picked. Are these optimal for some reason or were they simply found to be sufficient to get a predictive model? Are they expected to work for most optogenetic systems for which one might try to repeat the authors' methodology?

During our preliminary evaluation of the feasibility of the photoconversion-based model, we did perform some investigation to identify a set of characterization experiments sufficient for complete characterization of the model. First, we populated the model parameters with literature-based values (per their availability) and filled in the remaining parameters with best-guess values. Using this preliminary model, we constructed synthetic gene expression datasets corresponding to our proposed characterization experiments.

We initially planned just three characterization experiments (forward/reverse spectral measurements, and dark-to-active dynamics) based on both our intuition and our previous experience with optogenetic tools. We then added Gaussian noise to each measurement (assuming 10% uncertainty in each data point) and then used least-squares regression to fit the model to various noise realizations of the synthetic data. Given that we knew the model parameters used to generate the synthetic data, we were able to evaluate the capability of the experiments and regression procedure to reproduce these parameters.

With this approach, we found the use of just the three characterization experiments to be insufficient. In particular, we observed multicollinearity between the Hill 'k', the unit photoconversion rates ' $k_{1,i}$ ' and ' $k_{2,i}$ ', and the dark reversion rate ' k_{dr} '. That is, various realizations of these parameters could fit nearly equally well to the observed data. As we desired a more complete characterization of the systems, we set about designing an additional experiment in order to break the observed correlations between these parameters. Evaluation of the 'active-to-dark dynamics' experiment revealed that it was capable of breaking the relationship between these parameters.

At this point, we proceeded to collect the real-world experimental data and evaluate our approach. It appears that, despite our efforts, we were unable to fully break the multicollinearity between the stated dynamic parameters (as shown in Figure S5). Nonetheless, the model was fully capable of producing highly predictive results, and we proceeded with our demonstrations shown in Figures 3-7. Given the complexity of the description of this approach, and its ultimate failure to provide a complete characterization of the parameters, we concluded that it would be a distraction to include these details in the manuscript.

The question of optimal experiment design for efficient parameterization of optogenetic systems is indeed of great interest to us. We believe that given the demonstrated capabilities of the model, it ought to be possible to design such a set of nearly optimal experiments by performing a sensitivity analysis on the model itself. However, the clear caveat in this approach is that we indeed have no guarantee that other optogenetic systems will perform according to the model. While we certainly suspect that the underlying photoconversion mechanism will be conserved for most, if not all, optogenetic systems, the downstream signaling model which couples the light sensors to the system's output (gene expression, enzymatic activity, dimerization, protein localization, etc.) may require additional modeling efforts.

Ultimately, we hope that more optogenetic systems will be characterized to the same extent that we demonstrate here, in order to clarify and identify both the similarities and differences in the behavior exhibited by these systems. We hope that, given additional characterization data, a systematic classification scheme can be used to better capture the full diversity of optogenetic tools.

- In general it is difficult at times to follow what the different pieces of the model are. Sometimes they are called by different module names (i.e. "TCS signaling model") and other times as just "the Hill function". Also, the different k's get quite confusing, particularly when the k from the Hill function is mentioned.

We thank the reviewer for this opportunity to clarify the presentation of our manuscript. We have updated the text of the "Optogenetic TCS model" to more precisely identify the different sections of the model. We have also clarified the context of the Hill function within the TCS signaling portion of the output model in two locations in the Results ('Cph8-OmpR photoconversion model' and 'Development of a CcaSR, Cph8-OmpR dual-system model').

In addition, we have updated the main text in the introduction and in the first paragraph of the results to ensure that a description is provided in each instance a variable is introduced. We have also included additional cross-references to the parameter tables in Fig. 2f and Fig. S8f in the model sections of the Materials and Methods. Finally, we have replaced the Hill parameter 'k' with 'K' in order to more readily distinguish it from the various subscripted rate parameters. Taken together, we believe that these edits substantially improve the presentation and clarity of the manuscript, model, and variables.

Thank you again for sending us your revised manuscript. We have now heard back from reviewer #1 who was asked to evaluate your study and as you will see below, s/he is satisfied with the modifications made. I am therefore pleased to inform you that your paper has been accepted for publication.

REFEREE REPORT

Reviewer #1:

I am satisfied with the responses and revisions made by the authors.

Jeffrey J. Tabor

Molecular Systems Biology

MSB-16-7456